# Equipping Experts/Bandits with Long-term Memory

**Kai Zheng**[1,2]
zhengk92@pku.edu.cn

**Haipeng Luo**[3]
haipengl@usc.edu

**Ilias Diakonikolas**[4]
ilias.diakonikolas@gmail.com

**Liwei Wang**[1,2]
wanglw@cis.pku.edu.cn

## Abstract

We propose the first reduction-based approach to obtaining long-term memory guarantees for online learning in the sense of Bousquet and Warmuth [8], by reducing the problem to achieving typical switching regret. Specifically, for the classical expert problem with $K$ actions and $T$ rounds, using our framework we develop various algorithms with a regret bound of order $\mathcal{O}(\sqrt{T(S \ln T + n \ln K)})$ compared to any sequence of experts with $S-1$ switches among $n \leq \min\{S, K\}$ distinct experts. In addition, by plugging specific adaptive algorithms into our framework we also achieve the best of both stochastic and adversarial environments simultaneously. This resolves an open problem of Warmuth and Koolen [35]. Furthermore, we extend our results to the sparse multi-armed bandit setting and show both negative and positive results for long-term memory guarantees. As a side result, our lower bound also implies that sparse losses do not help improve the worst-case regret for contextual bandits, a sharp contrast with the non-contextual case.

## 1 Introduction

In this work, we propose a black-box reduction for obtaining long-term memory guarantees for two fundamental problems in online learning: the expert problem [17] and the multi-armed bandit (MAB) problem [6]. In both problems, a learner interacts with the environment for $T$ rounds, with $K$ fixed available actions. At each round, the environment decides the loss for each action while simultaneously the learner selects one of the actions and suffers the loss of this action. In the expert problem, the learner observes the loss of every action at the end of each round (a.k.a. full-information feedback), while in MAB, the learner only observes the loss of the selected action (a.k.a. bandit feedback).

For both problems, the classical performance measure is the learner's (static) regret, defined as the difference between the learner's total loss and the loss of the best fixed action. It is well-known that the minimax optimal regret is $\Theta(\sqrt{T \ln K})$ [17] and $\Theta(\sqrt{TK})$ [6, 4] for the expert problem and MAB respectively. Comparing against a fixed action, however, does not always lead to meaningful guarantees, especially when the environment is non-stationary and no single fixed action performs well. To address this issue, prior work has considered a stronger measure called switching/tracking/shifting regret, which is the difference between the learner's total loss and the loss

of a sequence of actions with at most $S-1$ switches. Various existing algorithms (including some black-box approaches) achieve the following switching regret

$$\begin{cases} \mathcal{O}(\sqrt{TS\ln(TK)}) & \text{for the expert problem [23, 21, 1, 27, 24],} \quad (1) \\ \mathcal{O}(\sqrt{TKS\ln(TK)}) & \text{for multi-armed bandits [6, 28].} \quad (2) \end{cases}$$

We call these *typical switching regret bounds*. Such bounds essentially imply that the learner pays the worst-case static regret for each switch in the benchmark sequence. While this makes sense in the worst case, intuitively one would hope to perform better if the benchmark sequence frequently switches back to previous actions, as long as the algorithm remembers which actions have performed well previously.

Indeed, for the expert problem, algorithms with long-term memory were developed that guarantee switching regret of order $\mathcal{O}\left(\sqrt{T(S\ln\frac{nT}{S} + n\ln\frac{K}{n})}\right)$, where $n \leq \min\{S, K\}$ is the number of distinct actions in the benchmark sequence [8, 2, 13].[1] Although there is no known lower bound, this regret bound essentially matches the one achieved by a computationally inefficient approach of running Hedge over all benchmark sequences with $S$ switches among $n$ experts, an approach that usually leads to the information-theoretically optimal regret guarantee. Compared to the typical switching regret bound of form (1) (which can be written as $\mathcal{O}(\sqrt{T(S\ln T + S\ln K)})$), this long-term memory guarantee implies that the learner pays the worst-case static regret only for each distinct action encountered in the benchmark sequence, and pays less for each switch, especially when $n$ is very small. Algorithms with long-term memory guarantees have been found to have better empirical performance [8], and applied to practical applications such as TCP round-trip time estimation [30], intrusion detection system [29], and multi-agent systems [31]. We are not aware of any similar studies for the bandit setting.

**Overview of our contributions.** The main contribution of this work is to propose a simple black-box approach to equip expert or MAB algorithms with long-term memory and to achieve switching regret guarantees of similar flavor to those of [8, 2, 13]. The key idea of our approach is to utilize a variant of the confidence-rated expert framework of [7], and to use a sub-routine to learn the confidence/importance of each action for each time. Importantly this sub-routine itself is an expert/bandit algorithm over *only two actions* and needs to enjoy some typical switching regret guarantee (for example of form (1) for the expert problem). In other words, our approach *reduces the problem of obtaining long-term memory to the well-studied problem of achieving typical switching regret*. Compared to existing methods [8, 2, 13], the advantages of our approach are the following:

1. While existing methods are all restricted to variants of the classical Hedge algorithm [17], our approach allows one to plug in a variety of existing algorithms and to obtain a range of different algorithms with switching regret $\mathcal{O}(\sqrt{T(S\ln T + n\ln K)})$. (Section 3.1)

2. Due to this flexibility, by plugging in specific adaptive algorithms, we develop a parameter-free algorithm whose switching regret is simultaneously $\mathcal{O}(\sqrt{T(S\ln T + n\ln K)})$ in the worst-case and $\mathcal{O}(S\ln T + n\ln(K\ln T))$ if the losses are piece-wise stochastic (see Section 2 for the formal definition). This is a generalization of previous best-of-both-worlds results for static or switching regret [19, 27], and resolves an open problem of Warmuth and Koolen [35]. The best previous bound for the stochastic case is $\mathcal{O}(S\ln(TK\ln T))$ [27]. (Section 3.2)

3. Our framework allows us to derive the first nontrivial long-term memory guarantees for the bandit setting, while existing approaches fail to do so (more discussion to follow). For example, when $n$ is a constant and the losses are sparse, our algorithm achieves switching regret $\mathcal{O}(S^{1/3}T^{2/3} + K^3\ln T)$ for MAB, which is better than the typical bound (2) when $S$ and $K$ are large. For example, when $S = \Theta(T^{\frac{7}{10}})$ and $K = \Theta(T^{\frac{3}{10}})$, our bound is of order $\mathcal{O}(T^{\frac{9}{10}}\ln T)$ while bound (2) becomes vacuous (linear in $T$), demonstrating a strict separation in learnability. (Section 4)

To motivate our results on long-term memory guarantees for MAB, a few remarks are in order. It is not hard to verify that existing approaches achieve switching regret $\mathcal{O}(\sqrt{TK(S\ln T + n\ln K)})$ for MAB. However, the polynomial dependence on the number of actions $K$ makes the improvement of

this bound over the typical bound (2) negligible. It is well-known that such polynomial dependence on $K$ is unavoidable in the worst-case due to the bandit feedback. This motivates us to consider situations where the necessary dependence on $K$ is much smaller. In particular, Bubeck et al. [10] recently showed that if the loss vectors are $\rho$-sparse, then a static regret bound of order $\mathcal{O}(\sqrt{T\rho \ln K} + K \ln T)$ is achievable, exhibiting a much more favorable dependence on $K$. We therefore focus on this sparse MAB problem and study what nontrivial switching regret bounds are achievable.

We first show that a bound of order $\mathcal{O}(\sqrt{T\rho S \ln(KT)} + KS \ln T)$, a natural generalization of the typical switching regret bound of (2) to the sparse setting, is impossible. In fact, we show that for any $S$ the worst-case switching regret is at least $\Omega(\sqrt{TKS})$, even when $\rho = 2$. Since achieving switching regret for MAB can be seen as a special case of contextual bandits [6, 26], this negative result also implies that, surprisingly, sparse losses do not help improve the worst-case regret for contextual bandits, which is a sharp contrast with the non-contextual case studied in [10] (see Theorem 6 and Corollary 7). Despite this negative result, however, as mentioned we are able to utilize our general framework to still obtain improvements over bound (2) when $n$ is small. Our construction is fairly sophisticated, requiring a special sub-routine that uses a novel *one-sided log-barrier regularizer* and admits a new kind of "local-norm" guarantee, which may be of independent interest.

## 2 Preliminaries

Throughout the paper, we use $[m]$ to denote the set $\{1, \ldots, m\}$ for some integer $m$. The learning protocol for the expert problem and MAB with $K$ actions and $T$ rounds is as follows: For each time $t = 1, \ldots, T$, (1) the learner first randomly selects an action $I_t \in [K]$ according to a distribution $p_t \in \Delta_K$ (the $(K-1)$-dimensional simplex); (2) simultaneously the environment decides the loss vector $\ell_t \in [-1, 1]^K$; (3) the learner suffers loss $\ell_t(I_t)$ and observes either $\ell_t$ in the expert problem (full-information feedback) or only $\ell_t(I_t)$ in MAB (bandit feedback). For any sequence of $T$ actions $i_1, \ldots, i_T \in [K]$, the expected regret of the learner against this sequence is defined as

$$\mathcal{R}(i_{1:T}) = \mathbb{E}\left[\sum_{t=1}^{T} \ell_t(I_t) - \sum_{t=1}^{T} \ell_t(i_t)\right] = \mathbb{E}\left[\sum_{t=1}^{T} r_t(i_t)\right],$$

where the expectation is with respect to both the learner and the environment and $r_t(i)$, the instantaneous regret (against action $i$), is defined as $p_t^\top \ell_t - \ell_t(i)$. When $i_1 = \cdots = i_T$, this becomes the traditional static regret against a fixed action. Most existing works on switching regret impose a constraint on the number of switches for the benchmark sequence: $\sum_{t=2}^{T} \mathbf{1}\{i_t \neq i_{t-1}\} \leq S - 1$. In other words, the sequence can be decomposed into $S$ disjoint intervals, each with a fixed comparator as in static regret. Typical switching regret bounds hold for any sequence with this constraint and are in terms of $T$, $K$ and $S$, such as Eq. (1) and Eq. (2).

The number of switches, however, does not fully characterize the difficulty of the problem. Intuitively, a sequence that frequently switches back to previous actions should be an easier benchmark for an algorithm with long-term memory that remembers which actions performed well in the past. To encode this intuition, prior works [8, 2, 13] introduced another parameter $n = |\{i_1, \ldots, i_T\}|$, the number of distinct actions in the sequence, to quantify the difficulty of the problem, and developed switching regret bounds in terms of $T$, $K$, $S$ and $n$. Clearly one has $n \leq \min\{S, K\}$, and we are especially interested in the case when $n \ll \min\{S, K\}$, which is natural if the data exhibits some periodic pattern. Our goal is to understand what improvements are achievable in this case and how to design algorithms that can leverage this property via a unified framework.

**Stochastic setting.** In general, we do not make any assumptions on how the losses are generated by the environment, which is known as the adversarial setting in the literature. We do, however, develop an algorithm (for the expert problem) that enjoys the best of both worlds — it not only enjoys some robust worst-case guarantee in the adversarial setting, but also achieves much smaller logarithmic regret in a stochastic setting. Specifically, in this stochastic setting, without loss of generality, we assume the $n$ distinct actions in $\{i_1, \ldots, i_T\}$ are $1, \ldots, n$. It is further assumed that for each $i \in [n]$, there exists a constant gap $\alpha_i > 0$ such that $\mathbb{E}_t[\ell_t(j) - \ell_t(i)] \geq \alpha_i$ for all $j \neq i$ and all $t$ such that $i_t = i$, where the expectation is with respect to the randomness of the environment conditioned on the history up to the beginning of round $t$. In other words, for every time step the algorithm is compared to the best action whose expected value is constant away from those of other actions. This is a natural generalization of the stochastic setting studied for static regret or typical switching regret [19, 27].

---
**Algorithm 1:** A Simple Reduction for Long-term Memory
---
**1** **Input**: expert algorithm $\mathcal{A}$ learning over $K$ actions with static regret guarantee (cf. Condition 1),
  expert algorithms $\mathcal{A}_1, \ldots, \mathcal{A}_K$ learning over two actions $\{0,1\}$ with switching regret guarantee (cf.
  Condition 2), parameter $\eta \leq 1/5$
**2** **for** $t = 1, 2, \ldots$ **do**
**3**     Receive sampling distribution $w_t \in \Delta_K$ from $\mathcal{A}$
**4**     Receive sampling probability $z_t(i)$ for action "1" from $\mathcal{A}_i$ for each $i \in [K]$
**5**     Sample $I_t \sim p_t$ where $p_t(i) \propto z_t(i)w_t(i)$, $\forall i$, and receive $\ell_t \in [-1,1]^K$
**6**     Feed loss vector $c_t$ to $\mathcal{A}$, where $c_t(i) = -z_t(i)r_t(i)$ with $r_t(i) = p_t^\top \ell_t - \ell_t(i)$
**7**     Feed loss vector $(0, 5\eta - r_t(i))$ to $\mathcal{A}_i$ for each $i \in [K]$
---

**Confidence-rated actions.** Our approach makes use of the confidence-rated expert setting of Blum and Mansour [7], a generalization of the expert problem (and the sleeping expert problem [18]). The protocol of this setting is the same as the expert problem, except that at the beginning of each round, the learner first receives a confidence score $z_t(i)$ for each action $i$. The regret against a fixed action $i$ is also scaled by its confidence and is now defined as $\mathbb{E}\left[\sum_{t=1}^T z_t(i)r_t(i)\right]$. The expert problem is clearly a special case with $z_t(i) = 1$ for all $t$ and $i$. There are a number of known examples showing why this formulation is useful, and our work will add one more to this list.

To obtain a bound on this new regret measure, one can in fact simply reduce it to the regular expert problem [7, 19, 27]. Specifically, let $\mathcal{A}$ be some expert algorithm over the same $K$ actions producing sampling distributions $w_1, \ldots, w_T \in \Delta_K$. The reduction works by sampling $I_t$ according to $p_t$ such that $p_t(i) \propto z_t(i)w_t(i)$, $\forall i$ and then feeding $c_t$ to $\mathcal{A}$ where $c_t(i) = -z_t(i)r_t(i)$, $\forall i$. Note that by the definition of $p_t$ one has $w_t^\top c_t = \sum_i w_t(i)z(i)(\ell_t(i) - p_t^\top \ell_t) = 0$. Therefore, one can directly equalize the confidence-rated regret and the regular static regret of the reduced problem: $\mathbb{E}\left[\sum_{t=1}^T z_t(i)r_t(i)\right] = \mathbb{E}\left[\sum_{t=1}^T (w_t^\top c_t - c_t(i))\right]$.

## 3 General Framework for the Expert Problem

In this section, we introduce our general framework to obtain long-term memory regret bounds and demonstrate how it leads to various new algorithms for the expert problem. We start with a simpler version and then move on to a more elaborate construction that is essential to obtain best-of-both-worlds results.

### 3.1 A simple approach for adversarial losses

A simple version of our approach is described in Algorithm 1. At a high level, it simply makes use of the confidence-rated action framework described in Section 2. The reduction to the standard expert problem is executed in Lines 5 and 6, with a black-box expert algorithm $\mathcal{A}$.

It remains to specify how to come up with the confidence score $z_t(i)$. We propose to *learn* these scores via a separate black-box expert algorithm $\mathcal{A}_i$ for each $i$. More specifically, each $\mathcal{A}_i$ is learning over two actions 0 and 1, where action 0 corresponds to confidence score 0 and action 1 corresponds to score 1. Therefore, the probability of picking action 1 at time $t$ naturally represents a confidence score between 0 and 1, which we denote by $z_t(i)$ overloading the notation (Line 4).

As for the losses fed to $\mathcal{A}_i$, we fix the loss of action 0 to be 0 (since shifting losses by the same amount has no real effect), and set the loss of action 1 to be $5\eta - r_t(i)$ (Line 7). The role of the term $-r_t(i)$ is intuitively clear — the larger the loss of action $i$ compared to the algorithm, the less confident we should be about it; the role of the constant bias term $5\eta$ will become clear in the analysis (in fact, it can even be removed at the cost of a worse bound — see Appendix B.2).

Finally we specify what properties we require from the black-box algorithms $\mathcal{A}, \mathcal{A}_1, \ldots, \mathcal{A}_K$. In short, $\mathcal{A}$ needs to ensure a static regret bound, while $\mathcal{A}_1, \ldots, \mathcal{A}_K$ need to ensure a switching regret bound. See Figure 1 for an illustration of our reduction. The trick is that since $\mathcal{A}_1, \ldots, \mathcal{A}_K$ are learning over only two actions, this construction helps us to separate the dependence on $K$ and the

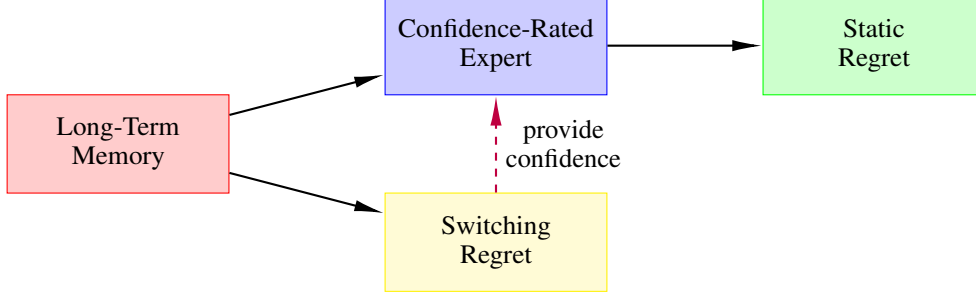

Figure 1: Illustration of reduction. The main idea of our approach is to reduce the problem of obtaining long-term memory guarantee to the confidence-rated expert problem and the problem of obtaining switching regret. Algorithms for the latter learn and provide confidence to the confidence-rated expert problem. The reduction from confidence-rated expert to obtaining standard static regret is known [7].

number of switches $S$. These (static or switching) regret bounds could be the standard worst-case $T$-dependent bounds mentioned in Section 1, in which case we would obtain looser long-term memory guarantees (specifically, $\sqrt{n}$ times worse — see Appendix B.2). Instead, we require these bounds to be data-dependent and in particular of the form specified below:

**Condition 1.** There exists a constant $C > 0$ such that for any $\eta \in (0, 1/5]$ and any loss sequence $c_1, \ldots, c_T \in [-2, 2]^K$, algorithm $\mathcal{A}$ (possibly with knowledge of $\eta$) produces sampling distributions $w_1, \ldots, w_T \in \Delta_K$ and ensures one of the following static regret bounds:

$$\sum_{t=1}^{T} w_t^\top c_t - \sum_{t=1}^{T} c_t(i) \leq \frac{C \ln K}{\eta} + \eta \sum_{t=1}^{T} |c_t(i)|, \quad \forall i \in [K] \tag{3}$$

$$\text{or} \quad \sum_{t=1}^{T} w_t^\top c_t - \sum_{t=1}^{T} c_t(i) \leq \frac{C \ln K}{\eta} + \eta \sum_{t=1}^{T} \left| w_t^\top c_t - c_t(i) \right|, \quad \forall i \in [K]. \tag{4}$$

**Condition 2.** There exists a constant $C > 0$ such that for any $\eta \in (0, 1/5]$, any loss sequence $h_1, \ldots, h_T \in [-3, 3]^2$, and any $S \in [T]$, algorithm $\mathcal{A}_i$ (possibly with knowledge of $\eta$) produces sampling distributions $q_1, \ldots, q_T \in \Delta_2$ and ensures one of the following switching regret bounds against any sequence $b_1, \ldots, b_T \in \{0, 1\}$ with $\sum_{t=2}^{T} \mathbf{1}\{b_t \neq b_{t-1}\} \leq S - 1$:[2]

$$\sum_{t=1}^{T} q_t^\top h_t - \sum_{t=1}^{T} h_t(b_t) \leq \frac{CS \ln T}{\eta} + \eta \sum_{t=1}^{T} |h_t(b_t)|, \tag{5}$$

$$\text{or} \quad \sum_{t=1}^{T} q_t^\top h_t - \sum_{t=1}^{T} h_t(b_t) \leq \frac{CS \ln T}{\eta} + \eta \sum_{t=1}^{T} \left| q_t^\top h_t - h_t(b_t) \right|, \tag{6}$$

$$\text{or} \quad \sum_{t=1}^{T} q_t^\top h_t - \sum_{t=1}^{T} h_t(b_t) \leq \frac{CS \ln T}{\eta} + \eta \sum_{t=1}^{T} \sum_{b \in \{0,1\}} q_t(b) |h_t(b)|. \tag{7}$$

We emphasize that these data-dependent bounds are all standard in the online learning literature,[3] and provide a few examples below (see Appendix A for brief proofs).

**Proposition 1.** *The following algorithms all satisfy Condition 1: Variants of Hedge [20, 34], Prod [12], Adapt-ML-Prod [19], AdaNormalHedge [27], and iProd/Squint [25].*

**Proposition 2.** *The following algorithms all satisfy Condition 2: Fixed-share [23], a variant of Fixed-share (Algorithm 5 in Appendix A), and AdaNormalHedge.TV [27].*

**Algorithm 2:** A Parameter-free Reduction for Best-of-both-worlds
---
1 **Define**: $M = \lfloor \log_2 \frac{\sqrt{T}}{5} \rfloor + 1$, $\eta_j = \min \left\{ \frac{1}{5}, \frac{2^{j-1}}{\sqrt{T}} \right\}$ for $j \in [M]$

2 **Input**: expert algorithm $\mathcal{A}$ learning over $KM$ actions with static regret guarantee (cf. Condition 3), expert algorithms $\{\mathcal{A}_{ij}\}_{i \in [K], j \in [M]}$ learning over two actions $\{0, 1\}$ with switching regret guarantee (cf. Condition 2)

3 **for** $t = 1, 2, \ldots$ **do**

4      Receive sampling distribution $w_t \in \Delta_{KM}$ from $\mathcal{A}$

5      Receive sampling probability $z_t(i, j)$ for action "1" from $\mathcal{A}_{ij}$ for each $i \in [K]$ and $j \in [M]$

6      Sample $I_t \sim p_t$ where $p_t(i) \propto \sum_{j=1}^{M} z_t(i,j) w_t(i,j)$, $\forall i$, and receive $\ell_t \in [-1, 1]^K$

7      Feed loss vector $c_t$ to $\mathcal{A}$, where $c_t(i, j) = -z_t(i, j) r_t(i)$ with $r_t(i) = p_t^\top \ell_t - \ell_t(i)$

8      Feed loss vector $(0, 5\eta_j |r_t(i)| - r_t(i))$ to $\mathcal{A}_{ij}$ for each $i \in [K]$ and $j \in [M]$
---

We are now ready to state the main result for Algorithm 1 (see Appendix B.1 for the proof).

**Theorem 3.** *Suppose Conditions 1 and 2 both hold. With $\eta = \min \left\{ \frac{1}{5}, \sqrt{\frac{S \ln T + n \ln K}{T}} \right\}$, Algorithm 1 ensures $\mathcal{R}(i_{1:T}) = \mathcal{O}\left( \sqrt{T(S \ln T + n \ln K)} \right)$ for any loss sequence $\ell_1, \ldots, \ell_T$ and benchmark sequence $i_1, \ldots, i_T$ such that $\sum_{t=2}^{T} \mathbf{1}\{i_t \neq i_{t-1}\} \leq S - 1$ and $|\{i_1, \ldots, i_T\}| \leq n$.*

Our bound in Theorem 3 is slightly worse than the existing bound of $\mathcal{O}\left( \sqrt{T(S \ln \frac{nT}{S} + n \ln \frac{K}{n})} \right)$ [8, 2],[4] but still improves over the typical switching regret $\mathcal{O}(\sqrt{T(S \ln T + S \ln K)})$ (Eq. (1)), especially when $n$ is small and $S$ and $K$ are large. To better understand the implication of our bounds, consider the following thought experiment. If the learner knew about the switch points (that is, $\{t : i_t \neq i_{t-1}\}$) that naturally divide the whole game into $S$ intervals, she could simply pick any algorithm with optimal static regret ($\sqrt{\text{"#rounds"} \ln K}$) and apply $S$ instances of this algorithm, one for each interval, which, via a direct application of the Cauchy-Schwarz inequality, leads to switching regret $\sqrt{TS \ln K}$. Compared to bound (1), this implies that the price of not knowing the switch points is $\sqrt{TS \ln T}$. Similarly, if the learner knew not only the switch points, but also the information on which intervals share the same competitor, then she could naturally apply $n$ instances of the static algorithm, one for each set of intervals with the same competitor. Again by the Cauchy-Schwarz inequality, this leads to switching regret $\sqrt{Tn \ln K}$. Therefore, our bound implies that the price of not having any prior information of the benchmark sequence is still $\sqrt{TS \ln T}$.

Compared to existing methods, our framework is more flexible and allows one to plug in any combination of the algorithms listed in Propositions 1 and 2. This flexibility is crucial and allows us to solve the problems discussed in the following sections. The approach of [2] makes use of a sleeping expert framework, a special case of the confidence-rated expert framework. However, their approach is not a general reduction and does not allow plugging in different algorithms. Finally, we note that our construction also shares some similarity with the black-box approach of [14] for a multi-task learning problem.

### 3.2 Best of both worlds

To further demonstrate the power of our approach, we now show how to use our framework to construct a parameter-free algorithm that enjoys the best of both adversarial and stochastic environments, resolving the open problem of [35] (see Algorithm 2). The key is to derive an adaptive switching regret bound that replaces the dependence on $T$ by the sum of the magnitudes of the instantaneous regret $\sum_t |r_t(i)|$, which previous works [19, 27] show is sufficient for adapting to the stochastic setting and achieving logarithmic regret.

To achieve this goal, the first modification we need is to change the bias term for the loss of action "1" for $\mathcal{A}_i$ from $5\eta$ to $5\eta|r_t(i)|$. Following the proof of Theorem 3, one can show that the dependence on $|\{t : i_t = i\}|$ now becomes $\sum_{t:i_t=i}|r_t(i)|$ for the regret against $i$. If we could tune $\eta$ optimally in terms of this data-dependent quality, then this would imply logarithmic regret in the stochastic setting by the same reasoning as in [19, 27].

However, the difficulty is that the optimal tuning of $\eta$ is unknown beforehand, and more importantly, different actions require tuning $\eta$ differently. To address this issue, at a high level we discretize the learning rate and pick $M = \Theta(\ln T)$ exponentially increasing values (Line 1), then we make $M = \Theta(\ln T)$ copies of each action $i \in [K]$, one for each learning rate $\eta_j$. More specifically, this means that the number of actions for $\mathcal{A}$ increases from $K$ to $KM$, and so does the number of sub-routines with switching regret, now denoted as $\mathcal{A}_{ij}$ for $i \in [K]$ and $j \in [M]$. Different copies of an action $i$ share the same loss $\ell_t(i)$ for $\mathcal{A}$, while action "1" for $\mathcal{A}_{ij}$ now suffers loss $5\eta_j|r_t(i)| - r_t(i)$ (Line 8). The rest of the construction remains the same. Note that selecting a copy of an action is the same as selecting the corresponding action, which explains the update rule of the sampling probability $p_t$ in Line 6 that marginalizes over $j$. Also note that for a vector in $\mathbb{R}^{KM}$ (e.g., $w_t, c_t, z_t$), we use $(i,j)$ to index its coordinates for $i \in [K]$ and $j \in [M]$.

Finally, with this new construction, we need algorithm $\mathcal{A}$ to exhibit a more adaptive static regret bound and in some sense be aware of the fact that different actions now correspond to different learning rates. More precisely, we replace Condition 1 with the following condition:

**Condition 3.** There exists a constant $C > 0$ such that for any $\eta_1, \ldots, \eta_M \in (0, 1/5]$ and any loss sequence $c_1, \ldots, c_T \in [-2, 2]^{KM}$, algorithm $\mathcal{A}$ (possibly with knowledge of $\eta_1, \ldots, \eta_M$) produces sampling distributions $w_1, \ldots, w_T \in \Delta_{KM}$ and ensures the following static regret bounds: for all $i \in [K]$ and $j \in [M]$:[5]

$$\sum_{t=1}^{T} w_t^\top c_t - \sum_{t=1}^{T} c_t(i,j) \leq \frac{C\ln(KM)}{\eta_j} + \eta_j \sum_{t=1}^{T} \left| w_t^\top c_t - c_t(i,j) \right|. \tag{8}$$

Once again, this requirement is achievable by many existing algorithms and we provide some examples below (see Appendix A for proofs).

**Proposition 4.** *The following algorithms all satisfy Condition 3: A variant of Hedge (Algorithm 6 in Appendix A), Adapt-ML-Prod [19], AdaNormalHedge [27], and iProd/Squint [25].*

We now state our main result for Algorithm 2 (see Appendix B.3 for the proof).

**Theorem 5.** *Suppose algorithm $\mathcal{A}$ satisfies Condition 3 and $\{\mathcal{A}_{ij}\}_{i\in[K],j\in[M]}$ all satisfy Condition 2. Algorithm 2 ensures that for any benchmark sequence $i_1, \ldots, i_T$ such that $\sum_{t=2}^{T} \mathbf{1}\{i_t \neq i_{t-1}\} \leq S - 1$ and $|\{i_1, \ldots, i_T\}| \leq n$, the following hold:*

- *In the adversarial setting, we have $\mathcal{R}(i_{1:T}) = \mathcal{O}\left(\sqrt{T(S\ln T + n\ln(K\ln T))}\right)$;*

- *In the stochastic setting (defined in Section 2), we have $\mathcal{R}(i_{1:T}) = \mathcal{O}\left(\sum_{i=1}^{n} \frac{S_i\ln T + \ln(K\ln T)}{\alpha_i}\right)$, where $S_i = 1 + \sum_{t=2}^{T} \mathbf{1}\left\{(i_{t-1} = i \wedge i_t \neq i) \vee (i_{t-1} \neq i \wedge i_t = i)\right\}$ s.t. $\sum_{i\in[n]} S_i \leq 3S$.[6]*

In other words, with a negligible price of $\ln\ln T$ for the adversarial setting, our algorithm achieves logarithmic regret in the stochastic setting with favorable dependence on $S$ and $n$. The best prior result is achieved by AdaNormalHedge.TV [27], with regret $\mathcal{O}\left(\sqrt{T(S\ln(TK\ln T))}\right)$ for the adversarial case and $\mathcal{O}\left(\sum_{i=1}^{n} \frac{S_i\ln(TK\ln T)}{\alpha_i}\right)$ for the stochastic case. We also remark that a variant of the algorithm of [8] with a doubling trick can achieve a guarantee similar to ours, but weaker in the sense that each $\alpha_i$ is replaced by $\min_i \alpha_i$. To the best of our knowledge this was previously unknown and we provide the details in Appendix B.4 for completeness.

**Algorithm 3:** A Sparse MAB Algorithm with Long-term Memory
---
**1** **Input**: parameter $\eta \leq \frac{1}{500}, \gamma, \delta$

**2** **Define**: regularizers $\psi(w) = \frac{1}{\eta} \sum_{i=1}^{K} w(i) \ln w(i) + \gamma \sum_{i=1}^{K} \ln \frac{1}{w(i)}$ and $\phi(z) = \frac{1}{\eta} \ln \frac{1}{z}$, Bregman divergence $D_\phi(z, z') = \phi(z) - \phi(z') - \phi'(z')(z - z')$

**3** **Initialize**: $w_1 = \frac{1}{K}$ where $\mathbf{1} \in \mathbb{R}^K$ is the all-one vector, and $z_1(i) = 1$ for all $i \in [K]$

**4** **for** $t = 1, 2, \ldots$ **do**

**5** $\quad$ Compute $\tilde{p}_t = (1 - \eta)p_t + \frac{\eta}{K}\mathbf{1}$ where $p_t(i) \propto z_t(i)w_t(i), \forall i$

**6** $\quad$ Sample $I_t \sim \tilde{p}_t$, receive $\ell_t(I_t)$, and construct loss estimator $\widehat{\ell}(i) = \frac{\ell_t(i)}{\tilde{p}_t(i)}\mathbf{1}\left\{i = I_t\right\}, \forall i$

**7** $\quad$ Set $r_t(i) = p_t^\top \widehat{\ell}_t - \widehat{\ell}_t(i)$ and $c_t(i) = -z_t(i)r_t(i) - \eta z_t(i)\widehat{\ell}_t(i)^2$ for each $i \in [K]$

**8** $\quad$ Update $w_{t+1} = \operatorname{argmin}_{w \in \Delta_K} \sum_{\tau=1}^{t} w^\top c_\tau + \psi(w)$ $\qquad\qquad\qquad$ ▷ update of $\mathcal{A}$

**9** $\quad$ Update $z_{t+1}(i) = \operatorname{argmin}_{z \in [\delta, 1]} -r_t(i)z + D_\phi(z, z_t(i))$ for each $i \in [K]$ $\qquad$ ▷ update of $\mathcal{A}_i$
---

## 4 Long-term Memory under Bandit Feedback

In this section, we move on to the bandit setting where the learner only observes the loss of the selected action $\ell_t(I_t)$ instead of $\ell_t$. As mentioned in Section 1, one could directly generalize the approach of [8, 2, 13] to obtain a bound of order $\mathcal{O}(\sqrt{TK(S \ln T + n \ln K)})$, a natural generalization of the full information guarantee, but such a bound is not a meaningful improvement compared to (2), due to the $\sqrt{K}$ dependence that is unavoidable for MAB in the worst case. Therefore, we consider a special case where the dependence on $K$ is much smaller: the sparse MAB problem [10]. Specifically, in this setting we make the additional assumption that all loss vectors are $\rho$-sparse for some $\rho \in [K]$, that is, $\|\ell_t\|_0 \leq \rho$ for all $t$. It was shown in [10] that for sparse MAB the static regret is of order $\mathcal{O}(\sqrt{T\rho \ln K} + K \ln T)$, exhibiting a much favorable dependence on $K$.

**Negative result.** To the best of our knowledge, there are no prior results on switching regret for sparse MAB. In light of bound (2), a natural conjecture would be that it would be possible to achieve switching regret of $\mathcal{O}(\sqrt{T\rho S \ln(KT)} + KS \ln T)$ with $S$ switches. Perhaps surprisingly, we show that this is in fact impossible.

**Theorem 6.** *For any $T, S, K \geq 2$ and any MAB algorithm, there exists a sequence of loss vectors that are 2-sparse, such that the switching regret of this algorithm is at least $\Omega(\sqrt{TKS})$.*

The high level idea of the proof is to force the algorithm to overfocus on one good action and thus miss an even better action later. This is similar to the construction of [15, Lemma 3] and [37, Theorem 4.1], and we defer the proof to Appendix C.1. This negative result implies that sparsity does not help improve the typical switching regret bound (2). In fact, since switching regret for MAB can be seen as a special case of the contextual bandits problem [6, 26], this result also immediately implies the following corollary, a sharp contrast compared to the positive result for the non-contextual case mentioned earlier (see Appendix C.1 for the definition of contextual bandit and related discussions).

**Corollary 7.** *Sparse losses do not help improve the worst-case regret for contextual bandits.*

**Long-term memory to the rescue.** Despite the above negative results, we next show how long-term memory can still help improve the switching regret for sparse MAB. Specifically, we use our general framework to develop a MAB algorithm whose switching regret is smaller than $\mathcal{O}(\sqrt{TKS})$ whenever $\rho$ and $n$ are small while $S$ and $K$ are large. Note that this is not a contradiction with Theorem 6, since in the construction of its proof, $n$ is as large as $\min\{S, K\}$.

At a high level, our algorithm (Algorithm 3) works by constructing the standard unbiased importance-weighted loss estimator $\widehat{\ell}_t$ (Line 6) and plugging it into our general framework (Algorithm 1). However, we emphasize that it is highly nontrivial to control the variance of these estimators without leading to bad dependence on $K$ in this framework where two types of sub-routines interact with each other. To address this issue, we design specialized sub-algorithms $\mathcal{A}$ and $\mathcal{A}_i$ to learn $w_t$ and $z_t(i)$ respectively. For learning $w_t$, we essentially deploy the algorithm of [10] for sparse MAB, which is an instance of the standard follow-the-regularized-leader algorithm with a special hybrid regularizer,

combining the entropy and the log-barrier (Lines 2 and 8). However, note that the loss $c_t$ we feed to this algorithm is *not sparse* and we cannot directly apply the guarantee from [10], but it turns out that one can still utilize the implicit exploration of this algorithm, as shown in our analysis. Compared to Algorithm 1, we also incorporate an extra bias term $-\eta z_t(i)\widehat{\ell}_t(i)^2$ in the definition of $c_t$ (Line 7), which is important for canceling the large variance of the loss estimator.

For learning $z_t(i)$ for each $i$, we design a new algorithm that is an instance of the standard Online Mirror Descent algorithm (see e.g., [22]). Recall that this is a one-dimensional problem, as we are trying to learn the distribution $(1 - z_t(i), z_t(i))$ over actions $\{0, 1\}$. We design a special one-dimensional regularizer $\phi(z) = \frac{1}{\eta} \ln \frac{1}{z}$, which can be seen as a *one-sided log-barrier*,[7] to bias towards action "1". Technically, this provides a special "local-norm" guarantee that is critical for our analysis and may be of independent interest (see Lemma 14 in Appendix C.2). In addition, we remove the bias term in the loss for action "1" (so it is only $-r_t(i)$ now) as it does not help in the bandit case, and we also force $z_t(i)$ to be at least $\delta$ for some parameter $\delta$, which is important for achieving switching regret. Line 9 summarizes the update for $z_t(i)$.

Finally, we also enforce a small amount of uniform exploration by sampling $I_t$ from $\tilde{p}_t$, a smoothed version of $p_t$ (Line 5). We present the main result of our algorithm below (proven in Appendix C.2).

**Theorem 8.** *With $\eta = \max\left\{S^{\frac{1}{3}}\rho^{-\frac{2}{3}}(nT)^{-\frac{1}{3}}, \sqrt{\frac{\ln K}{T\rho}}\right\}, \delta = \sqrt{\frac{S}{T\eta n}}, \gamma = 200K^2$, Algorithm 3 ensures*

$$\mathcal{R}(i_{1:T}) = \mathcal{O}\left((\rho S)^{\frac{1}{3}}(nT)^{\frac{2}{3}} + n\sqrt{T\rho \ln K} + nK^3 \ln T\right) \qquad (9)$$

*for any sequence of $\rho$-sparse losses $\ell_1, \ldots, \ell_T$ and any benchmark sequence $i_1, \ldots, i_T$ such that $\sum_{t=2}^{T} \mathbf{1}\{i_t \neq i_{t-1}\} \leq S - 1$ and $|\{i_1, \ldots, i_T\}| \leq n$.*

In the case when $\rho$ and $n$ are constants, our bound (9) becomes $\mathcal{O}(S^{\frac{1}{3}}T^{\frac{2}{3}} + K^3 \ln T)$, which improves over the existing bound $\mathcal{O}(\sqrt{TKS \ln(TK)})$ when $(\frac{T}{S})^{\frac{1}{3}} < K < (TS)^{\frac{1}{5}}$ (also recall the example in Section 1 where our bound is sublinear in $T$ while existing bounds become vacuous).

As a final remark, one might wonder if similar best-of-both-worlds results are also possible for MAB in terms of switching regret, given the positive results for static regret [9, 33, 5, 32, 36, 38]. We point out that the answer is negative — the proof of [37, Theorem 4.1] implicitly implies that even with one switch, logarithmic regret is impossible for MAB in the stochastic setting.

## 5  Conclusion

In this work, we propose a simple reduction-based approach to obtaining long-term memory regret guarantee. By plugging various existing algorithms into this framework, we not only obtain new algorithms for this problem in the adversarial case, but also resolve the open problem of Warmuth and Koolen [35] that asks for a single algorithm achieving the best of both stochastic and adversarial environments in this setup. We also extend our results to the bandit setting and show both negative and positive results.

One clear open question is whether our bound for the bandit case (Theorem 8) can be improved, and more generally what is the best achievable bound in this case.

**Acknowledgments.**  The authors would like to thank Alekh Agarwal, Sébastien Bubeck, Dylan Foster, Wouter Koolen, Manfred Warmuth, and Chen-Yu Wei for helpful discussions. Kai Zheng and Liwei Wang were supported by Natioanl Key R&D Program of China (no. 2018YFB1402600), BJNSF (L172037). Haipeng Luo was supported by NSF Grant IIS-1755781. Ilias Diakonikolas was supported by NSF Award CCF-1652862 (CAREER) and a Sloan Research Fellowship.

## Footnotes

[1] Key Laboratory of Machine Perception, MOE, School of EECS, Peking University

[2] Center for Data Science, Peking University

[3] University of Southern California

[4] University of Wisconsin-Madison

[1] The setting considered in [8, 2] is in fact slightly different from, yet closely related to, the expert problem. One can easily translate their regret bounds into the bounds we present here.

[2]In terms of notation in Algorithm 1, $q_t = (1 - z_t(i), z_t(i))$.

[3]In fact, most standard bounds replace the absolute value we present here with square, leading to even smaller bounds (up to a constant). We choose to use the looser ones with absolute values since this makes the conditions weaker while still being sufficient for all of our analysis.

[4]In fact, using the adaptive guarantees of AdaNormalHedge [27] or iProd/Squint [25] that replaces the $\ln K$ dependence in Eq. (4) by a KL divergence term, one can further improve the term $n \ln K$ in our bound to $n \ln \frac{K}{n}$ matching previous bounds. Since this improvement is small, we omit the details.

[5]In fact an analogue of Eq. (3) with individual learning rates would also suffice, but we are not aware of any algorithms that achieve such guarantee.

[6]This definition of $S_i$ is the same as the one in the proof of Theorem 3.

[7]The usual log-barrier regularizer (see e.g. [16, 3, 36]) would be $\frac{1}{\eta}(\ln \frac{1}{z} + \ln \frac{1}{1-z})$ in this case.

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
