[Supplementary Material · Appendix-Equipping Experts-Bandits with Long-term Memory.pdf]

## A  Examples of Sub-algorithms

In this section, we briefly discuss why the algorithms listed in Propositions 1, 2, and 4 satisfy Conditions 1, 2, and 3 respectively. We first note that except for AdaNormalHedge [27], all other algorithms satisfy even tighter bounds with the absolute value replaced by square (also see Footnote 3).

### A.1  Condition 1

Prod [12] with learning rate $\eta$ satisfies Eq. (4) according to its original analysis. Adapt-ML-Prod [19], AdaNormalHedge [27], and iProd/Squint [25] are all parameter-free algorithms that satisfy for all $i \in [K]$,

$$\sum_{t=1}^{T} w_t^\top c_t - c_t(i) \leq \mathcal{O}\left(\sqrt{(\ln K) \sum_{t=1}^{T} \left| w_t^\top c_t - c_t(i) \right|} + \ln K\right). \tag{10}$$

By AM-GM inequality the square root term can be upper bounded by $\frac{\ln K}{4\eta} + \eta \sum_{t=1}^{T} \left| w_t^\top c_t - c_t(i) \right|$ for any $\eta$. Also the constraint $\eta \leq 1/5$ in Condition 1 allows one to bound the extra $\ln K$ term by $\frac{\ln K}{5\eta}$. This leads to Eq. (4).

Finally, for completeness we present a variant of Hedge (Algorithm 4) that can be extracted from [20, 31] and that satisfies Eq. (3).

**Proposition 9.** *Algorithm 4 satisfies Eq.* (3).

*Proof.* Define $\Phi_t = \sum_{i=1}^{K} \exp\left(\eta R_t(i) - \eta^2 G_t(i)\right)$ where $R_t(i) = \sum_{\tau=1}^{t} r_\tau(i)$ with $r_\tau(i) = w_\tau^\top c_\tau - c_\tau(i)$ and $G_t(i) = \sum_{\tau=1}^{t} c_\tau^2(i)$. The goal is to show $\Phi_T \leq \Phi_{T-1} \leq \cdots \leq \Phi_0 = K$, which implies for any $i$, $\exp\left(\eta R_T(i) - \eta^2 G_T(i)\right) \leq K$ and thus Eq. (3) after rearranging. Indeed, for any $t$ we have

$$\Phi_t - \Phi_{t-1}$$
$$= \sum_i \exp\left(\eta R_{t-1}(i) - \eta^2 G_{t-1}(i)\right) \left(\exp\left(\eta r_t(i) - \eta^2 c_t^2(i)\right) - 1\right)$$
$$= \exp\left(\eta w_t^\top c_t\right) \sum_i \exp\left(\eta R_{t-1}(i) - \eta^2 G_{t-1}(i)\right) \left(\exp\left(-\eta c_t(i) - \eta^2 c_t^2(i)\right) - \exp\left(-\eta w_t^\top c_t\right)\right)$$
$$\leq \exp\left(\eta w_t^\top c_t\right) \sum_i \exp\left(\eta R_{t-1}(i) - \eta^2 G_{t-1}(i)\right) \left(1 - \eta c_t(i) - \exp\left(-\eta w_t^\top c_t\right)\right)$$
$$\leq \exp\left(\eta w_t^\top c_t\right) \sum_i \exp\left(\eta R_{t-1}(i) - \eta^2 G_{t-1}(i)\right) \eta r_t(i)$$
$$= 0,$$

where the first inequality uses the fact $\exp(x - x^2) \leq 1 + x$ for any $x \geq -1/2$, the second inequality uses the fact $-\exp(-x) \leq x - 1$ for any $x$, and the last equality holds since $w_t(i) \propto \exp\left(\eta R_{t-1}(i) - \eta^2 G_{t-1}(i)\right)$ and $\sum_i w_t(i) r_t(i) = 0$. $\qquad\square$

### A.2  Condition 2

We first note that the three algorithms we include in Proposition 2 all work for an arbitrary number of actions $K$ (instead of just two actions) and the general guarantee will be in the same form of Eq. (5), (6), and (7) except that $\ln T$ is replaced by $\ln(KT)$.

Fixed-share [23] with learning rate $\eta$ satisfies Eq. (7) and the proof can be extracted from the proof of [6, Theorem 8.1] or [28, Theorem 2]. AdaNormalHedge.TV [27] is again a parameter-free algorithm and achieves the bound of (6) using similar tricks mentioned earlier for Condition 1.

Finally we provide a variant of Fixed-share that satisfies Eq. (5). The pseudocode is in Algorithm 5, where we adopt the notation from Condition 2 ($q_t$ for distribution, $h_t$ for loss, $b$ for action index) but present the general case with $K$ actions.

**Proposition 10.** *Algorithm 5 satisfies Eq.* (5).

**Algorithm 4:** Hedge Variant 1

1 **Input**: learning rate $\eta \in (0, 1/5]$
2 **for** $t = 1, 2, \ldots$ **do**
3      Sample $I_t \sim w_t$ where $w_t(i) \propto \exp\left(-\sum_{\tau < t}(\eta c_\tau(i) + \eta^2 c_\tau^2(i))\right)$
4      Receive loss $c_t \in [-1, 1]^K$

---

**Algorithm 5:** Fixed-share Variant

1 **Input**: learning rate $\eta \in (0, 1/5], \gamma = 1/T$
2 **Initialize**: $\tilde{q}_1 = \frac{1}{K}$
3 **for** $t = 1, 2, \ldots$ **do**
4      Sample an action according to $q_t = (1 - \gamma)\tilde{q}_t + \frac{\gamma}{K}\mathbf{1}$
5      Receive loss $h_t \in [-1, 1]^K$
6      Compute $\tilde{q}_{t+1}$ such that $\tilde{q}_{t+1}(b) \propto q_t(b) \exp(-\eta h_t(b) - \eta^2 h_t^2(b))$

---

*Proof.* We first write the algorithm as an instance of Online Mirror Descent. Let $\psi(q) = \sum_{b=1}^{K} q(b) \ln q(b)$ be the entropy regularizer, and $\bar{q}_{t+1}$ be such that $\nabla\psi(\bar{q}_{t+1}) = \nabla\psi(q_t) - \eta h_t - \eta^2 h_t^2$ where $h_t^2$ represents the element-wise square. Then one can verify $\bar{q}_{t+1}(b) = q_t(b) \exp(-\eta h_t(b) - \eta^2 h_t^2(b))$ and $\tilde{q}_{t+1} = \operatorname{argmin}_{q \in \Delta_K} D_\psi(q, \bar{q}_{t+1})$, where $D_\psi(q, q') = \sum_b \left( q(b) \ln \frac{q(b)}{q'(b)} + q'(b) - q(b) \right)$ is the Bregman divergence associated with $\psi$. Now we have for any $q \in \Delta_K$,

$$
\begin{aligned}
&\langle q_t - q, \eta h_t + \eta^2 h_t^2 \rangle \\
&= \langle q_t - q, \nabla\psi(q_t) - \nabla\psi(\bar{q}_{t+1}) \rangle \\
&= D_\psi(q, q_t) - D_\psi(q, \bar{q}_{t+1}) + D_\psi(q_t, \bar{q}_{t+1}) \\
&\leq D_\psi(q, q_t) - D_\psi(q, \tilde{q}_{t+1}) + D_\psi(q_t, \bar{q}_{t+1}) \\
&= D_\psi(q, q_t) - D_\psi(q, \tilde{q}_{t+1}) + \sum_{b=1}^{K} q_t(b)\left(\eta h_t(b) + \eta^2 h_t^2(b) + \exp(-\eta h_t(b) - \eta^2 h_t^2(b)) - 1\right) \\
&\leq D_\psi(q, q_t) - D_\psi(q, \tilde{q}_{t+1}) + \eta^2 \sum_{b=1}^{K} q_t(b) h_t^2(b) \\
&\leq D_\psi(q, q_t) - D_\psi(q, q_{t+1}) + 2\gamma + \eta^2 \sum_{b=1}^{K} q_t(b) h_t^2(b),
\end{aligned}
$$

where the first inequality is by the generalized Pythagorean theorem, the second inequality is by the fact $\exp(x - x^2) \leq 1 + x$ for all $x \geq -1/2$, and the last one is by the definition of $q_{t+1}$ and the fact $\ln\frac{1}{1-\gamma} \leq 2\gamma$ for any $\gamma \leq 1/2$. Rearranging then gives

$$
\langle q_t - q, h_t \rangle \leq \frac{D_\psi(q, q_t) - D_\psi(q, q_{t+1}) + 2\gamma}{\eta} + \eta \sum_{b=1}^{K} q(b) h_t^2(b).
$$

A benchmark sequence with $S - 1$ switches naturally divides the sequence into $S$ intervals, and for each interval $1 \leq s, \ldots, e \leq T$, by summing up the inequality above from $t = s$ to $t = e$ and telescoping we have

$$
\begin{aligned}
\sum_{t=s}^{e} \langle q_t - q, h_t \rangle &\leq \frac{D_\psi(q, q_s) + 2(t - s + 1)\gamma}{\eta} + \eta \sum_{t=s}^{e} \sum_{b=1}^{K} q(b) h_t^2(b) \\
&\leq \frac{\ln\frac{K}{\gamma} + 2(t - s + 1)\gamma}{\eta} + \eta \sum_{t=s}^{e} \sum_{b=1}^{K} q(b) h_t^2(b).
\end{aligned}
$$

Finally summing over all intervals, setting $q$ to put all weight on the corresponding competitor, and realizing $\gamma = 1/T$ finish the proof. $\qquad\square$

---
**Algorithm 6:** Hedge Variant 2
---
1 **Input**: learning rate $\eta_1, \ldots, \eta_K \in (0, 1/5]$
2 **for** $t = 1, 2, \ldots$ **do**
3     Sample $I_t \sim w_t$ where $w_t(i) \propto \eta_i \exp\left(\sum_{\tau < t}(\eta_i r_\tau(i) - \eta_i^2 r_\tau^2(i))\right), r_\tau(i) = w_\tau^\top c_\tau - c_\tau(i)$
4     Receive loss $c_t \in [-1, 1]^K$
---

## A.3 Condition 3

To simplify notation, we use $K$ to denote the number of actions (instead of $KM$) and prove the following

$$\sum_{t=1}^{T} w_t^\top c_t - c_t(i) \leq \frac{C \ln K}{\eta_i} + \eta_i \sum_{t=1}^{T} \left| w_t^\top c_t - c_t(i) \right|. \tag{11}$$

which clearly implies Eq. (8). Once again since Adapt-ML-Prod [19], AdaNormalHedge [27], and iProd/Squint [25] are all parameter-free algorithms satisfying Eq. (10), they also ensure Eq. (11) for any $\eta_i \leq 1/5$ by the same reasoning mentioned for Condition 1. Next we present a variant of Hedge (Algorithm 6) with individual learning rate for each action and prove the following.

**Proposition 11.** *Algorithm 6 satisfies Eq. (11).*

*Proof.* Define $\Phi_{t,i} = \exp\left(\sum_{\tau=1}^{t}(\eta_i r_\tau(i) + \eta_i^2 r_\tau^2(i))\right)$. We have

$$
\begin{aligned}
\ln\left(\sum_{i=1}^{K} \Phi_{t,i}\right) - \ln\left(\sum_{i=1}^{K} \Phi_{t-1,i}\right) &= \ln \frac{\sum_i \Phi_{t-1,i} e^{\eta_i r_t(i) - \eta_i^2 r_t^2(i)}}{\sum_i \Phi_{t-1,i}} \\
&\leqslant \ln \frac{\sum_i \Phi_{t-1,i}(1 + \eta_i r_t(i))}{\sum_i \Phi_{t-1,i}} \\
&= \ln \frac{\sum_i \Phi_{t-1,i}}{\sum_i \Phi_{t-1,i}} = 0,
\end{aligned}
$$

where the inequality holds by the fact $\exp(x - x^2) \leqslant 1 + x$ for any $x \geq -1/2$ and the equality holds because $w_t(i) \propto \eta_i \Phi_{t-1,i}$ and $\sum_i w_t(i) r_t(i) = 0$. Therefore,

$$\ln K = \ln \sum_i \Phi_{0,i} \geq \cdots \geq \ln \sum_i \Phi_{T,i} \geq \ln \Phi_{T,i} = \sum_{t=1}^{T}(\eta_i r_t(i) - \eta_i^2 r_t^2(i)).$$

Solving for $\sum_t r_t(i)$ then proves Eq. (11). $\qquad\square$

# B   Proofs for Section 3

In this section we provide proofs and related discussions for our algorithms under full-information feedback (i.e. the expert problem).

## B.1   Proof of Theorem 3

*Proof.* For each distinct action $i$ in $\mathcal{J} = \{i_1, \ldots, i_T\}$, we first apply the static regret bound of $\mathcal{A}$ stated in Condition 1 (either Eq. (3) or Eq. (4)). With the fact $w_t^\top c_t = 0$ and $|r_t(i)| \leq 2$ this gives

$$\sum_{t=1}^{T} z_t(i) r_t(i) \leq \frac{C \ln K}{\eta} + 2\eta \sum_{t=1}^{T} z_t(i). \tag{12}$$

Next we apply the switching regret bound of $\mathcal{A}_i$ stated in Condition 2 with $b_t = 0$ if $i_t \neq i$ and $b_t = 1$ otherwise (note that $q_t = (1 - z_t(i), z_t(i))$ and $h_t = (0, 5\eta - r_t(i))$). This gives with

437  $S_i = 1 + \sum_{t=2}^{T} \mathbf{1}\{b_t \neq b_{t-1}\}$ and $T_i = |\{t : i_t = i\}|$

$$\sum_{t=1}^{T} z_t(i)(5\eta - r_t(i)) \leq \sum_{t:i_t=i} (5\eta - r_t(i)) + \frac{CS_i \ln T}{\eta} + \eta B, \tag{13}$$

438  where $B$ is

$$\begin{cases} \sum_{t:i_t=i} |5\eta - r_t(i)| & \text{if Eq. (5) holds,} \\ \sum_{t:i_t \neq i} z_t(i)|5\eta - r_t(i)| + \sum_{t:i_t=i}(1 - z_t(i))|5\eta - r_t(i)| & \text{if Eq. (6) holds,} \\ \sum_{t=1}^{T} z_t(i)|5\eta - r_t(i)| & \text{if Eq. (7) holds.} \end{cases}$$

439  In either case, using the fact $|5\eta - r_t(i)| \leq 3$ we have

$$B \leq 3\sum_{t=1}^{T} z_t(i) + 3T_i.$$

440  Combining this inequality with Eq. (13) and rearranging give

$$\sum_{t:i_t=i} r_t(i) \leq \frac{CS_i \ln T}{\eta} + 8\eta T_i + \sum_{t=1}^{T} \left( z_t(i)r_t(i) - 2\eta z_t(i) \right). \tag{14}$$

441  Further combining inequalities (12) and (14) and canceling terms give

$$\sum_{t:i_t=i} r_t(i) \leq \frac{C(S_i \ln T + \ln K)}{\eta} + 8\eta T_i. \tag{15}$$

442  Finally summing over $i \in \mathcal{J}$, using the fact $\mathcal{R}(i_{1:T}) = \mathbb{E}\left[\sum_{i\in\mathcal{J}} \sum_{t:i_t=i} r_t(i)\right]$, $\sum_{i\in\mathcal{J}} S_i \leq$
443  $2S + n \leq 3S$, $\sum_{i\in\mathcal{J}} T_i = T$, $|\mathcal{J}| \leq n$ and the choice of $\eta$ finish the proof. $\qquad\square$

## B.2  A weaker bound via weaker conditions

445  Condition 1 and Condition 2 require some data-dependent regret bounds. In fact, one can even relax
446  these conditions and replace the data-dependent regret bounds with worst-case $T$-dependent bounds,
447  leading to a slightly weaker long-term memory guarantee. Specifically, if we replace the bounds in
448  Condition 1 and Condition 2 by standard worst-case static and switching regret bounds

$$\sum_{t=1}^{T} w_t^\top c_t - c_t(i) = \mathcal{O}\left(\sqrt{T \ln K}\right) \quad \text{and} \quad \sum_{t=1}^{T} q_t^\top h_t - h_t(b_t) = \mathcal{O}\left(\sqrt{TS \ln T}\right)$$

449  respectively, then by setting $\eta = 0$ in Algorithm 1 (that is, removing the bias term in the loss for $\mathcal{A}_i$)
450  and redoing the proof of Theorem 3 in a similar way one can verify that Eq. (15) now becomes

$$\sum_{t:i_t=i} r_t(i) = \mathcal{O}\left(\sqrt{T(S_i \ln T + \ln K)}\right),$$

451  which finally leads to

$$\mathcal{R}(i_{1:T}) = \mathcal{O}\left(\sqrt{T(nS \ln T + n^2 \ln K)}\right)$$

452  via Cauchy-Schwarz inequality. Compared to our bound in Theorem 3, this leads to an extra $\sqrt{n}$
453  factor.

## B.3  Proof of Theorem 5

455  *Proof.* The first step is to prove that for each distinct action $i \in \mathcal{J} = \{i_1, \ldots, i_T\}$, Algorithm 2
456  ensures

$$\sum_{t:i_t=i} r_t(i) \leq \mathcal{O}\left(\sqrt{(S_i \ln T + \ln(KM))\mathbb{E}\left[\sum_{t:i_t=i} |r_t(i)|\right]} + S_i \ln T + \ln(KM)\right). \tag{16}$$

The proof is similar to that of Theorem 3. We first apply the static regret bound of $\mathcal{A}$ stated in Condition 3, which gives for any $i \in [K]$ and $j \in [M]$,

$$\sum_{t=1}^{T} z_t(i,j) r_t(i) \leq \frac{C \ln(KM)}{\eta_j} + \eta_j \sum_{t=1}^{T} z_t(i,j) |r_t(i)|. \tag{17}$$

Here we use the fact

$$w_t^\top c_t = -\sum_{i,j} w_t(i,j) z_t(i,j) r_t(i) = -\mathcal{Z} \sum_i p_t(i) r_t(i) = -\mathcal{Z} \left( p_t^\top \ell_t - \sum_i p_t(i) \ell_t(i) \right) = 0$$

where $\mathcal{Z} = \sum_{i,j} w_t(i,j) z_t(i,j)$ is the normalization factor. Next we apply the switching regret bound of $\mathcal{A}_{ij}$ stated in Condition 2 with $\eta = \eta_j$, $b_t = 0$ if $i_t \neq i$ and $b_t = 1$ otherwise (note that $q_t = (1 - z_t(i,j), z_t(i,j))$ and $h_t = (0, 5\eta_j |r_t(i)| - r_t(i)))$. This gives with $S_i = 1 + \sum_{t=2}^{T} \mathbf{1}\{b_t \neq b_{t-1}\}$,

$$\sum_{t=1}^{T} z_t(i,j)(5\eta_j |r_t(i)| - r_t(i)) \leq \sum_{t:i_t=i} (5\eta_j |r_t(i)| - r_t(i)) + \frac{C S_i \ln T}{\eta_j} + \eta_j B, \tag{18}$$

where $B$ is

$$\begin{cases} \sum_{t:i_t=i} |5\eta_j |r_t(i)| - r_t(i)| & \text{if Eq. (5) holds,} \\ \sum_{t:i_t \neq i} z_t(i,j)|5\eta_j |r_t(i)| - r_t(i)| + \sum_{t:i_t=i}(1 - z_t(i,j))|5\eta_j |r_t(i)| - r_t(i)| & \text{if Eq. (6) holds,} \\ \sum_{t=1}^{T} z_t(i,j)|5\eta_j |r_t(i)| - r_t(i)| & \text{if Eq. (7) holds.} \end{cases}$$

In either case, using the fact $5\eta_j \leq 1$ and thus $|5\eta_j |r_t(i)| - r_t(i)| \leq 2|r_t(i)|$, we have

$$B \leq 2\sum_{t=1}^{T} z_t(i,j)|r_t(i)| + 2\sum_{t:i_t=i} |r_t(i)|.$$

Combining this inequality with Eq. (18) and rearranging give

$$\sum_{t:i_t=i} r_t(i) \leq \frac{C S_i \ln T}{\eta_j} + 7\eta_j \sum_{t:i_t=i} |r_t(i)| + \sum_{t=1}^{T} (z_t(i,j)r_t(i) - 3\eta_j z_t(i,j)|r_t(i)|). \tag{19}$$

Further combining inequalities (17) and (19) and canceling terms give

$$\sum_{t:i_t=i} r_t(i) \leq \frac{C(S_i \ln T + \ln(KM))}{\eta_j} + 7\eta_j \sum_{t:i_t=i} |r_t(i)|.$$

Now we pick $j$ such that

$$\eta_j \leq \min\left\{1/5, \sqrt{\frac{S_i \ln T + \ln(KM)}{\sum_{t:i_t=i} |r_t(i)|}}\right\} \leq 2\eta_j,$$

which is always possible by the construction of $\eta_1, \ldots, \eta_M$. This proves Eq. (16).

**Adversarial setting.** We simply bound $|r_t(i)|$ by 2 in Eq. (16). The rest is the same as the proof of Theorem 3: summing over $i \in \mathcal{J}$, applying Cauchy-Schwarz inequality, and using the fact $\mathcal{R}(i_{1:T}) = \mathbb{E}\left[\sum_{i \in \mathcal{J}} \sum_{t:i_t=i} r_t(i)\right]$, $\sum_{i \in \mathcal{J}} S_i \leq 2S + n \leq 3S$, $\sum_{i \in \mathcal{J}} \sum_{t:i_t=i} 1 = T$, $|\mathcal{J}| \leq n$, $M = \Theta(\ln T)$ prove $\mathcal{R}(i_{1:T}) = \mathcal{O}\left(\sqrt{T(S \ln T + n \ln(K \ln T))}\right)$.

**Stochastic setting.** The proof is similar to that of [27] and solely replies on the adaptive bound (16). Recall that in the stochastic setting, without loss of generality we assume $\{i_1, \ldots, i_T\} = [n]$. For each $i \in [n]$ there exists a constant gap $\alpha_i$ such that $\mathbb{E}_t[\ell_t(j) - \ell_t(i)] \geq \alpha_i$ for all $j \neq i$ and all $t$ such that $i_t = i$. This implies

$$\mathbb{E}\left[\sum_{t:i_t=i} r_t(i)\right] = \mathbb{E}\left[\sum_{t:i_t=i} \sum_{j \neq i} p_t(j)(\ell_t(j) - \ell_t(i))\right]$$

$$\geq \alpha_i \mathbb{E}\left[\sum_{t:i_t=i}\sum_{j\neq i}p_t(j)\right] = \alpha_i \mathbb{E}\left[\sum_{t:i_t=i}(1-p_t(i))\right].$$

On the other hand, we have

$$\sum_{t:i_t=i}|r_t(i)| = \sum_{t:i_t=i}\left|\sum_{j\neq i}p_t(j)(\ell_t(j)-\ell_t(i))\right| \leq \sum_{t:i_t=i}\sum_{j\neq i}p_t(j)|(\ell_t(j)-\ell_t(i))| \leq 2\sum_{t:i_t=i}(1-p_t(i)).$$

Combining the two inequalities above with Eq. (16) and by AM-GM inequality, we know that there exists a constant $C'$ such that

$$\alpha_i \mathbb{E}\left[\sum_{t:i_t=i}(1-p_t(i))\right] \leq \mathbb{E}\left[\sum_{t:i_t=i}r_t(i)\right] \leq \frac{C'(S_i\ln T + \ln(KM))}{\alpha_i} + \frac{\alpha_i}{2}\mathbb{E}\left[\sum_{t:i_t=i}(1-p_t(i))\right].$$

Rearranging proves

$$\frac{\alpha_i}{2}\mathbb{E}\left[\sum_{t:i_t=i}(1-p_t(i))\right] \leq \frac{C'(S_i\ln T + \ln(KM))}{\alpha_i}$$

and thus

$$\mathbb{E}\left[\sum_{t:i_t=i}r_t(i)\right] \leq \frac{2C'(S_i\ln T + \ln(KM))}{\alpha_i}.$$

Summing over $i \in [n]$ finishes the proof. $\qquad\square$

### B.4 A weaker best-of-both-worlds result

In this section we present a version of the "Mixing Past Posteriors" algorithm of [8, 2, 13] with a particular doubling trick and show that it also provides some similar but weaker best-of-both-worlds results. As far as we know this is unknown previously.

The pseudocode is in Algorithm 7. It is a variant of Hedge where each time the sampling distribution mixes all the past distributions. We apply a standard doubling trick to the quantity $\sum_t \sum_i p_t(i)r_t^2(i)$, an important data-dependent quantity that turns out to be useful for adapting to the stochastic setting (similar to the role of $\sum_t \sum_i |r_t(i)|$ in Eq. (16)). Specifically the algorithm satisfies the following adaptive switching regret bound.

**Theorem 12.** *Algorithm 7 ensures*

$$\mathcal{R}(i_{1:T}) = \mathcal{O}\left(\sqrt{(S\ln T + n\ln K)\sum_{t=1}^{T}\sum_{i=1}^{K}p_t(i)r_t^2(i)}\right), \tag{20}$$

*for any loss sequence $\ell_1, \ldots, \ell_T$ and benchmark sequence $i_1, \ldots, i_T$ such that $\sum_{t=2}^{T}\mathbf{1}\{i_t \neq i_{t-1}\} \leq S - 1$ and $|\{i_1, \ldots, i_T\}| \leq n$. This implies that*

- *in the adversarial setting, we have $\mathcal{R}(i_{1:T}) = \mathcal{O}\left(\sqrt{T(S\ln T + n\ln K)}\right)$;*

- *in the stochastic setting (defined in Section 2), we have $\mathcal{R}(i_{1:T}) = \mathcal{O}\left(\frac{S\ln T + n\ln K}{\min_{i\in[n]}\alpha_i}\right)$.*

Compared to our bounds in Theorem 5, one can see that the stochastic bound here is weaker in the sense that all $\alpha_i$'s are replaced by $\min_i \alpha_i$. At a technical level, this is because this algorithm only admits an adaptive regret bound (20) over the entire horizon, instead of a bound like Eq. (16) that holds over segments with the same competitor.

*Proof.* Similar to the proof of Proposition 10, we start by writing the algorithm as an instance of Online Mirror Descent. Let $\psi(p) = \sum_{i=1}^{K}p(i)\ln p(i)$ be the entropy regularizer, and $\bar{p}_{t+1}$ be such that $\nabla\psi(\bar{p}_{t+1}) = \nabla\psi(p_t) + \eta r_t$. Then one can verify $\bar{p}_{t+1}(i) = p_t(i)\exp(\eta r_t(i))$ and

---

**Algorithm 7:** Mixing Past Posteriors with Doubling Trick

---

1 **Initialize:** $\gamma = 1/T, V = 0, t_0 = 1, D = 1, \eta = \min\left\{1/5, \sqrt{(S \ln T + n \ln K)/D}\right\}, \tilde{p}_1 = \frac{1}{K}$

2 **for** $t = 1, 2, \ldots$ **do**

3     Sample an action according to $p_t = (1 - \gamma)\tilde{p}_t + \frac{\gamma}{t - t_0} \sum_{\tau = t_0}^{t-1} \tilde{p}_\tau$

4     Receive loss $\ell_t \in [-1, 1]^K$

5     Update $\tilde{p}_{t+1}$ such that $\tilde{p}_{t+1}(i) \propto p_t(i) \exp(\eta r_t(i))$, where $r_t(i) = p_t^\top \ell_t - \ell_t(i)$

6     Update $V \leftarrow V + \sum_{i=1}^K p_t(i) r_t^2(i)$

7     **if** $V > D$ **then**                                          ▷ restart condition

8        Set $V = 0, t_0 = t + 1, D \leftarrow 2D, \eta = \min\left\{1/5, \sqrt{(S \ln T + n \ln K)/D}\right\}, \tilde{p}_{t+1} = \frac{1}{K}$

---

505 $\tilde{p}_{t+1} = \text{argmin}_{p \in \Delta_K} D_\psi(p, \bar{p}_{t+1})$, where $D_\psi(p, p') = \sum_i \left(p(i) \ln \frac{p(i)}{p'(i)} + p'(i) - p(i)\right)$ is the

506 Bregman divergence associated with $\psi$. Now we have for any $p \in \Delta_K$, we have

$$
\begin{aligned}
\langle p, \eta r_t \rangle &= \langle p_t - p, -\eta r_t \rangle && (\langle p_t, r_t \rangle = 0) \\
&= \langle p_t - p, \nabla \psi(p_t) - \nabla \psi(\bar{p}_{t+1}) \rangle \\
&= D_\psi(p, p_t) - D_\psi(p, \bar{p}_{t+1}) + D_\psi(p_t, \bar{p}_{t+1}) \\
&\leq D_\psi(p, p_t) - D_\psi(p, \tilde{p}_{t+1}) + D_\psi(p_t, \bar{p}_{t+1}) && \text{(generalized Pythagorean theorem)} \\
&= D_\psi(p, p_t) - D_\psi(p, \tilde{p}_{t+1}) + \sum_{i=1}^K p_t(i) \left(-\eta r_t(i) + \exp(\eta r_t(i)) - 1\right) \\
&\leq D_\psi(p, p_t) - D_\psi(p, \tilde{p}_{t+1}) + \eta^2 \sum_{i=1}^K p_t(i) r_t^2(i). && (e^x - 1 \leq x + x^2, \forall x < 1/2)
\end{aligned}
$$

507 Now consider a period between two resets of the algorithm that starts at time $t_0$ and ends at time $t_1$.
508 Let $s_t = 1 + \max\{t_0 \leq s < t : i_s = i_t\}$ be one plus the most recent time when $i_t$ is the competitor
509 (if the set is empty, $s_t$ is defined as 1). Note that by the definition of $p_t$ we have

$$
D_\psi(p, p_t) = \sum_i p(i) \ln \frac{p(i)}{p_t(i)} \leq D_\psi(p, \tilde{p}_{s_t}) + \mathbf{1}\{s_t = t\} \ln \frac{1}{1 - \gamma} + \mathbf{1}\{s_t \neq t\} \ln \frac{T}{\gamma}.
$$

510 Therefore, combining previous bounds we have for any $j \in [K]$,

$$
r_t(j) \leq \frac{\ln \frac{\tilde{p}_{t+1}(j)}{\tilde{p}_{s_t}(j)} + \mathbf{1}\{s_t = t\} \ln \frac{1}{1-\gamma} + \mathbf{1}\{s_t \neq t\} \ln \frac{T}{\gamma}}{\eta} + \eta \sum_{i=1}^K p_t(i) r_t^2(i).
$$

511 Summing over $t$ in this period and telescoping lead to

$$
\begin{aligned}
\sum_{t=t_0}^{t_1} r_t(i_t) &\leq \frac{n \ln K + T \ln \frac{1}{1-\gamma} + S \ln \frac{T}{\gamma}}{\eta} + \eta \sum_{t=t_0}^{t_1} \sum_{i=1}^K p_t(i) r_t^2(i) \\
&\leq \frac{\mathcal{O}(n \ln K + S \ln T)}{\eta} + \eta \sum_{t=t_0}^{t_1} \sum_{i=1}^K p_t(i) r_t^2(i) && \text{(by the choice of } \gamma) \\
&\leq \frac{\mathcal{O}(n \ln K + S \ln T)}{\eta} + \eta(D + 1) && \text{(by the restart condition)} \\
&\leq \mathcal{O}(\sqrt{(n \ln K + S \ln T)D} + n \ln K + S \ln T) && \text{(by the choice of } \eta)
\end{aligned}
$$

512 Finally suppose there are $k = \mathcal{O}(\ln T)$ periods in total, then

$$
\mathcal{R}(i_{1:T}) = \mathcal{O}\left(\sqrt{(n \ln K + S \ln T) 2^k} + (n \ln K + S \ln T) \ln T\right).
$$

513 Note that in this case by the restart condition one must also have $\sum_{t=1}^T \sum_{i=1}^K p_t(i) r_t^2(i) \geq 2^{k-1}$,
514 which implies Eq. (20) (by dropping the lower order term $(n \ln K + S \ln T) \ln T$ for simplicity).

**Adversarial setting.** Simply upper bound $\sum_{t=1}^{T}\sum_{i=1}^{K} p_t(i)r_t^2(i)$ by $4T$.

**Stochastic setting.** This is similar to the proof of Theorem 5. We make the following two observations. First, by the definition of the stochastic setting we have

$$
\mathcal{R}(i_{1:T}) = \mathbb{E}\left[\sum_{i\in[n]}\sum_{t:i_t=i} r_t(i)\right] = \mathbb{E}\left[\sum_{i\in[n]}\sum_{t:i_t=i}\sum_{j\neq i} p_t(j)(\ell_t(j)-\ell_t(i))\right]
$$

$$
\geq \sum_{i\in[n]}\alpha_i\mathbb{E}\left[\sum_{t:i_t=i}\sum_{j\neq i} p_t(j)\right] \geq \left(\min_{i\in[n]}\alpha_i\right)\mathbb{E}\left[\sum_{t=1}^{T}(1-p_t(i_t))\right].
$$

On the other hand, we have $r_t^2(i_t) \leq 2|\sum_{i\neq i_t} p_t(i)(\ell_t(i)-\ell_t(i_t))| \leq 4(1-p_t(i_t))$ and thus

$$
\sum_{i=1}^{K} p_t(i)r_t^2(i) = p_t(i_t)r_t^2(i_t) + \sum_{i\neq i_t} p_t(i)r_t^2(i)
$$

$$
\leqslant 4p_t(i_t)(1-p_t(i_t)) + 4(1-p_t(i_t))
$$

$$
\leqslant 8(1-p_t(i_t)).
$$

Combining the two inequalities above with Eq. (20) and by AM-GM inequality, we know that there exists a constant $C'$ such that

$$
\left(\min_{i\in[n]}\alpha_i\right)\mathbb{E}\left[\sum_{t=1}^{T}(1-p_t(i_t))\right] \leq \mathcal{R}(i_{1:T}) \leq \frac{C'(S\ln T + \ln K)}{\min_{i\in[n]}\alpha_i} + \frac{\min_{i\in[n]}\alpha_i}{2}\mathbb{E}\left[\sum_{t=1}^{T}(1-p_t(i_t))\right].
$$

Rearranging proves

$$
\frac{\min_{i\in[n]}\alpha_i}{2}\mathbb{E}\left[\sum_{t=1}^{T}(1-p_t(i_t))\right] \leq \frac{C'(S\ln T + \ln K)}{\min_{i\in[n]}\alpha_i}
$$

and thus the claimed regret bound. $\square$

# C   Proofs for Section 4

In this section we provide the omitted proofs for Section 4.

## C.1   Negative results

*Proof of Theorem 6.* Divide the whole horizon evenly into $S/2$ intervals. Our goal is to show that for any algorithm $\mathcal{A}$, there exists a sequence of 2-sparse loss vectors such that the switching regret of $\mathcal{A}$ against a benchmark with at most 2 switches on each of these intervals is at least $\Omega(\sqrt{TK/S})$, this clearly implies that the overall switching regret against a benchmark with at most $S$ switches is at least $\Omega(\sqrt{TKS})$.

To show this, consider a fixed interval and consider the behavior of $\mathcal{A}$ against a fixed loss vector $-\frac{1}{2}e_1$ for the entire interval ($e_i$ represents a basis vector). Let $\mathcal{N}$ be the expected number of times that action 1 is not selected by $\mathcal{A}$ on this interval (a fixed number conditioned on everything prior to this interval). If $\mathcal{N} \geq \sqrt{TK/S}$, then the (static) regret of $\mathcal{A}$ against action 1 on this interval is already $\Omega(\sqrt{TK/S})$. Otherwise, there must exist an action $i \neq 1$ such that in expectation it is selected for less than $\frac{\sqrt{TK/S}}{K-1} \leq 2\sqrt{T/(KS)}$ times. In this case, there must also exist a subinterval of length $\frac{2T/S}{4\sqrt{T/(KS)}} = \frac{1}{2}\sqrt{TK/S}$ where in expectation action $i$ is selected for less than $1/2$ times. This means that with probability at least $1/2$, action $i$ is not selected at all on this subinterval. If we switch the loss vector from $-\frac{1}{2}e_1$ to $-\frac{1}{2}e_1 - e_i$ starting from the beginning of this subinterval, $\mathcal{A}$ suffers expected regret $\Omega(\sqrt{TK/S})$ against action $i$ after the switch point. In other words, in this case the switching regret of $\mathcal{A}$ (first against 1 and then against $i$) is $\Omega(\sqrt{TK/S})$, finishing the proof. $\square$

To prove Corollary 7, we first remind the reader the contextual bandit setting [6, 26]. It is a generalization of the MAB problem where at the beginning of each round $t$, the learner first observes a *context* $x_t$ from some arbitrary context space $\mathcal{X}$, and then selects an action $I_t$ and observes its loss $\ell_t(I_t)$. The learner is given a fixed set of *policies* $\Pi$ beforehand where each policy is a mapping from $\mathcal{X}$ to $[K]$. The (static) regret of the learner against a fixed policy $\pi \in \Pi$ is now defined as

$$\mathcal{R}(\pi) = \mathbb{E}\left[\sum_{t=1}^{T} \ell_t(I_t) - \ell_t(\pi(x_t))\right].$$

The optimal regret for a finite policy class $\Pi$ is known to be $\Theta(\sqrt{TK \ln |\Pi|})$.

It is well-known that one can reduce the problem of achieving switching regret (with $S$ switches) for MAB to the problem of achieving static regret for contextual bandit. To do this, simply let $x_t = t$ and $\Pi$ be the set of action sequences with length $T$ and $S$ switches. For a policy $\pi$ that corresponds to the action sequence $i_1, \ldots, i_T$, its output at time $t$ is simply $\pi(x_t) = i_t$. Comparing the regret definitions it is clear that the static regret for this contextual bandit problem exactly corresponds to the switching regret for MAB. Moreover, since the size of $\Pi$ in this case is $\mathcal{O}((TK)^S)$, a static regret of form $\Theta(\sqrt{TK \ln |\Pi|})$ exactly recovers the typical switching regret bound of form (2). Now it is clear that Corollary 7 is directly implied by Theorem 6.

## C.2   Proof of Theorem 8

The proof relies on the following two lemmas, which respectively state the static and switching regret guarantees for algorithm $\mathcal{A}$ (that learns $w_t$) and algorithm $\mathcal{A}_i$ (that learns $z_t(i)$).

**Lemma 13.** *With $\gamma = 200K^2$, Algorithm 3 ensures for any $i \in [K]$,*

$$\mathbb{E}\left[\sum_{t=1}^{T} w_t^\top c_t - \sum_{t=1}^{T} c_t(i)\right] \leq \mathcal{O}\left(T\rho\eta + \frac{\ln K}{\eta} + K^3 \ln T\right)$$

**Lemma 14.** *For any $i \in [K]$, Line 9 of Algorithm 3 ensures*

$$-\sum_{t=1}^{T} z_t(i)r_t(i) + \sum_{t=1}^{T} u_t r_t(i) \leq \eta \sum_{t=1}^{T} z_t(i)r_t^2(i) + \frac{2S_i}{\eta\delta} \tag{21}$$

*for any sequence of $r_1(i), \ldots, r_T(i) \in \mathbb{R}$ and any competitor sequence $u_1, \ldots, u_T \in [\delta, 1]$ with $\sum_{t=2}^{T} \mathbf{1}\{u_t \neq u_{t-1}\} \leq S_i - 1$.*

The bound in Lemma 13 resembles the one of [10] for sparse MAB, but as mentioned since $c_t$ is not sparse (nor can it be made sparse after shifting), it requires a different analysis. The bound in Lemma 14 contains a "local-norm" term $\sum_{t=1}^{T} z_t(i)r_t^2(i)$ that resembles the one achieved by Hedge in the full information setting. However, importantly this holds for *any real-valued sequence* of $r_1(i), \ldots, r_T(i)$, while Hedge requires the losses to be bounded from one side. We are not able to prove the same bound with the usual log barrier regularizer (see Footnote 7) either. As far as we know this lemma is new and might be of independent interest.

Combining these two lemmas we now provide the proof for Theorem 8, followed by the proofs of these lemmas.

*Proof of Theorem 8.* First note that by the definition of $c_t, r_t$ and $p_t$ one has

$$w_t^\top c_t = \sum_{i=1}^{K} -w_t(i)z_t(i)r_t(i) - \eta w_t(i)z_t(i)\widehat{\ell}_t^2(i)$$

$$= -\eta \sum_{i=1}^{K} w_t(i)z_t(i)\widehat{\ell}_t^2(i).$$

For each distinct action $i \in \mathcal{J} = \{i_1, \ldots, i_T\}$, applying Lemma 13 and rearranging then lead to

$$\sum_{t=1}^{T} \mathbb{E}\left[z_t(i)r_t(i) + \eta z_t(i)\widehat{\ell}_t^2(i)\right] \leq \eta\mathbb{E}\left[\sum_{t=1}^{T}\sum_{j=1}^{K} w_t(j)z_t(j)\widehat{\ell}_t^2(j)\right] + \mathcal{O}\left(T\rho\eta + \frac{\ln K}{\eta} + K^3 \ln T\right).$$

$$\tag{22}$$

Next we apply Lemma 14 by setting $u_t = \delta$ if $i_t \neq i$ and $u_t = 1$ otherwise, which gives

$$\sum_{t:i_t=i} r_t(i) \leq -\delta \sum_{t:i_t \neq i} r_t(i) + \sum_{t=1}^{T} z_t(i) r_t(i) + \eta \sum_{t=1}^{T} z_t(i) r_t^2(i) + \frac{2S_i}{\eta\delta}. \quad (23)$$

Let $\mathbb{E}_t$ denote the expectation conditioned on the history up to the beginning of round $t$. It is clear that $\widehat{\ell}_t$ is unbiased: $\mathbb{E}_t[\widehat{\ell}_t] = \ell_t$, and thus $\mathbb{E}_t[-r_t(i)] = \ell_t(i) - p_t^\top \ell_t(i) \leq 2$. Also we have

$$\begin{aligned}
r_t^2(i) &= \left(p_t^\top \widehat{\ell}_t\right)^2 - 2\left(p_t^\top \widehat{\ell}_t\right)\widehat{\ell}_t(i) + \widehat{\ell}_t^2(i) \\
&= \left(\frac{p_t(I_t)\ell_t(I_t)}{\widetilde{p}_t(I_t)}\right)^2 - 2\left(\frac{p_t(I_t)}{\widetilde{p}_t(I_t)}\right)\ell_t(I_t)\widehat{\ell}_t(i) + \widehat{\ell}_t^2(i) \\
&\leq \left(\frac{p_t(I_t)}{\widetilde{p}_t(I_t)}\right)^2 + \widehat{\ell}_t^2(i) \\
&\leq \left(\frac{1}{1-\eta}\right)^2 + \widehat{\ell}_t^2(i) \leq 4 + \widehat{\ell}_t^2(i),
\end{aligned} \quad (24)$$

where the first inequality uses the fact $\ell_t(I_t)\widehat{\ell}_t(i) \geq 0$ (since it is either $0$ or $\ell_t(i)^2/\widetilde{p}_t(i)$), the second inequality uses the definition of $\widetilde{p}_t$, and the last one uses $\eta \leq 1/2$. Combining these with Eq. (22) and Eq. (23) gives

$$\mathbb{E}\left[\sum_{t:i_t=i} r_t(i)\right] \leq 2T\delta + \frac{2S_i}{\eta\delta} + \eta\mathbb{E}\left[\sum_{t=1}^{T}\sum_{j=1}^{K} w_t(j)z_t(j)\widehat{\ell}_t^2(j)\right] + \mathcal{O}\left(T\rho\eta + \frac{\ln K}{\eta} + K^3 \ln T\right).$$

It remains to bound

$$\begin{aligned}
\mathbb{E}_t\left[\sum_{j=1}^{K} w_t(j)z_t(j)\widehat{\ell}_t^2(j)\right] &= \sum_{j=1}^{K} w_t(j)z_t(j)\frac{\ell_t^2(j)}{\widetilde{p}_t(j)} \\
&\leq 2\sum_{j=1}^{K} w_t(j)z_t(j)\frac{\ell_t^2(j)}{p_t(j)} \\
&\leq 2\sum_{j=1}^{K} \ell_t^2(j) \leq 2\rho,
\end{aligned}$$

which implies

$$\mathbb{E}\left[\sum_{t:i_t=i} r_t(i)\right] \leq 2T\delta + \frac{2S_i}{\eta\delta} + \mathcal{O}\left(T\rho\eta + \frac{\ln K}{\eta} + K^3 \ln T\right).$$

Summing over $i \in \mathcal{J}$ and using the fact $\sum_{i \in \mathcal{J}} S_i \leq 3S$ and $\mathcal{R}(i_{1:T}) \leq \mathbb{E}\left[\sum_{t=1}^{T} r_t(i_t)\right] + T\eta$ give

$$\mathcal{R}(i_{1:T}) = \mathcal{O}\left(nT\delta + \frac{S}{\eta\delta} + nT\rho\eta + \frac{n \ln K}{\eta} + nK^3 \ln T\right).$$

Plugging in the parameters $\eta$ and $\delta$ proves the theorem. □

*Proof of Lemma 13.* The proof is in similar spirit of those of [10, 11]. Define for a semi-definite matrix $M$ the associated norm for a vector $a$ as $\|a\|_M = \sqrt{a^\top M a}$. By standard analysis of Follow-the-Regularized-Leader, we have for any $w \in \Delta_K$,

$$\sum_{t=1}^{T}(w_t - w)^\top c_t \leq \mathcal{O}\left(\sum_{t=1}^{T} \|c_t\|_{\nabla^{-2}\psi(w_t')}^2 + D_\psi(w, w_1)\right),$$

where $w'_t$ is some point on the segment connecting $w_t$ and $w_{t+1}$, and $D_\psi$ is the Bregman divergence associated with $\psi$. Set $w = (1 - \frac{1}{T})e_i + \frac{1}{TK}\mathbf{1}$. One can verify $\mathbb{E}\left[\sum_{t=1}^{T} w^\top c_t - c_t(i)\right] = \mathcal{O}(K)$ and $D_\psi(w, w_1) = \frac{\ln K}{\eta} + \gamma K \ln T$, and thus

$$\mathbb{E}\left[\sum_{t=1}^{T} w_t^\top c_t - c_t(i)\right] = \mathcal{O}\left(\mathbb{E}\left[\sum_{t=1}^{T} \|c_t\|^2_{\nabla^{-2}\psi(w'_t)}\right] + \frac{\ln K}{\eta} + \gamma K \ln T\right).$$

The rest of the proof consists of two steps. First, we prove that the algorithm is stable in the sense that $\frac{1}{2} \leq \frac{w_{t+1}(i)}{w_t(i)} \leq 2$ for all $t$ and $i$, which implies $\|c_t\|^2_{\nabla^{-2}\psi(w'_t)} = \mathcal{O}\left(\|c_t\|^2_{\nabla^{-2}\psi(w_t)}\right)$. The second step is to show $\mathbb{E}_t\left[\|c_t\|^2_{\nabla^{-2}\psi(w_t)}\right] = \mathcal{O}(\rho\eta)$. Combining these two steps finishes the proof.

**First step.** To prove the stability, it suffices to show $\|w_t - w_{t+1}\|_{\nabla^2\psi(w_t)} \leq \frac{1}{2}$. Indeed, this is because $\nabla^2\psi(w_t) \succcurlyeq \gamma\left[\frac{1}{w_t(i)^2}\right]_{\text{diag}}$, where $\left[\frac{1}{w_t(i)^2}\right]_{\text{diag}}$ represents the $K$ dimensional diagonal matrix whose $i$-th diagonal element is $\frac{1}{w_t(i)^2}$, and thus $\|w_t - w_{t+1}\|_{\nabla^2\psi(w_t)} \leq \frac{1}{2}$ implies $\|w_t - w_{t+1}\|_{\gamma[1/w_t(i)^2]_{\text{diag}}} \leq \frac{1}{2}$, which further implies $1 - \frac{1}{2\sqrt{\gamma}} \leqslant \frac{w_{t+1}(i)}{w_t(i)} \leqslant 1 + \frac{1}{2\sqrt{\gamma}}$ and thus $\frac{1}{2} \leq \frac{w_{t+1}(i)}{w_t(i)} \leq 2$.

To prove $\|w_t - w_{t+1}\|_{\nabla^2\psi(w_t)} \leq \frac{1}{2}$, define $F_t(w) = \sum_{s=1}^{t} w_s^\top c_s + \psi(w)$ so that $w_{t+1} = \text{argmin}_{w \in \Delta_K} F_t(w)$. We will prove $F_t(w') \geq F_t(w_t)$ for any $w'$ such that $\|w' - w_t\|_{\nabla^2\psi(w_t)} = \frac{1}{2}$, which then implies $\|w_t - w_{t+1}\|_{\nabla^2\psi(w_t)} \leq \frac{1}{2}$ by the convexity of $F_t$.

Indeed, by Taylor's expansion, there exists some $\xi$ on the line segment joining $w'$ and $w_t$, such that

$$F_t(w') = F_t(w_t) + \nabla F_t(w_t)^\top (w' - w_t) + \frac{1}{2}(w' - w_t)^\top \nabla^2 F_t(\xi)(w' - w_t)$$

$$= F_t(w_t) + c_t^\top(w' - w_t) + \nabla F_{t-1}(w_t)^\top(w' - w_t) + \frac{1}{2}\|w' - w_t\|^2_{\nabla^2\psi(\xi)}$$

$$\geq F_t(w_t) + c_t^\top(w' - w_t) + \frac{1}{2}\|w' - w_t\|^2_{\nabla^2\psi(\xi)}$$

$$\geq F_t(w_t) - \|c_t\|_{\nabla^{-2}\psi(w_t)}\|w' - w_t\|_{\nabla^2\psi(w_t)} + \frac{1}{2}\|w' - w_t\|^2_{\nabla^2\psi(\xi)}$$

$$= F_t(w_t) - \frac{1}{2}\|c_t\|_{\nabla^{-2}\psi(w_t)} + \frac{1}{2}\|w' - w_t\|^2_{\nabla^2\psi(\xi)}$$

where the first inequality is by the first order optimality of $w_t$ and the second is by Hölder's inequality. Note that $\xi$ is between $w_t$ and $w'$, which implies $\|\xi - w_t\|_{\nabla^2\psi(w_t)} \leq \frac{1}{2}$ and $\frac{\xi_i}{w_t(i)} \leq 1 + \frac{1}{2\sqrt{\gamma}} \leq \frac{11}{10}$ similar to previous discussions. Therefore, we have $\nabla^2\psi(\xi) \succcurlyeq \frac{100}{121}\nabla^2\psi(w_t)$, and thus

$$F_t(w') \geq F_t(w_t) - \frac{1}{2}\|c_t\|_{\nabla^{-2}\psi(w_t)} + \frac{50}{121}\|w' - w_t\|^2_{\nabla^2\psi(w_t)} = F_t(w_t) - \frac{1}{2}\|c_t\|_{\nabla^{-2}\psi(w_t)} + \frac{25}{242}.$$

Next we show $\|c_t\|^2_{\nabla^{-2}\psi(w_t)} \leq \frac{1}{25}$, which will finish the proof for the stability.

$$\|c_t\|^2_{\nabla^{-2}\psi(w_t)} = \sum_{i=1}^{K} \frac{\eta w_t^2(i)}{w_t(i) + \gamma\eta}c_t^2(i)$$

$$\leq 2\sum_{i=1}^{K} \frac{\eta w_t^2(i)}{w_t(i) + \gamma\eta}\left(z_t^2(i)r_t^2(i) + \eta^2 z_t^2(i)\widehat{\ell}_t^4(i)\right) \qquad \text{(Cauchy-Schwarz)}$$

$$\leq 2\sum_{i=1}^{K} \frac{\eta w_t^2(i)}{w_t(i) + \gamma\eta}\left(4z_t^2(i) + z_t^2(i)\widehat{\ell}_t^2(i) + \eta^2 z_t^2(i)\widehat{\ell}_t^4(i)\right) \qquad \text{(by Eq. (24))}$$

$$\leq 8\eta\sum_i w_t(i)z_t^2(i) + \frac{2}{\gamma}\sum_i w_t^2(i)z_t^2(i)\widehat{\ell}_t^2(i) + \frac{2\eta^2}{\gamma}\sum_i w_t^2(i)z_t^2(i)\widehat{\ell}_t^4(i)$$

$$\leq 8\eta + \frac{2p_t^2(I_t)}{\gamma\tilde{p}_t^2(I_t)} + \frac{2\eta^2 p_t^2(I_t)}{\gamma\tilde{p}_t^4(I_t)} \qquad\qquad \text{(by definition of } \widehat{\ell}_t)$$

$$\leq 8\eta + \frac{2}{\gamma(1-\eta)^2} + \frac{2\eta^2}{\gamma(1-\eta)^2} \cdot \frac{K^2}{\eta^2} \qquad\qquad \text{(by definition of } \tilde{p}_t)$$

$$\leqslant \frac{1}{25}. \qquad\qquad \text{(by } \eta \leq \tfrac{1}{500} \text{ and } \gamma = 200K^2)$$

**Second step.** With the stability, it is clear that $\|c_t\|^2_{\nabla^{-2}\psi(w'_t)} = \mathcal{O}\left(\|c_t\|^2_{\nabla^{-2}\psi(w_t)}\right)$. Now we show $\mathbb{E}_t\left[\|c_t\|^2_{\nabla^{-2}\psi(w_t)}\right] = \mathcal{O}(\rho\eta)$. Note that this is similar to previous calculations, but the expectation allows us to bound the term by something even smaller. Specifically, we continue from the intermediate step of the previous calculation

$$\|c_t\|^2_{\nabla^{-2}\psi(w_t)} \leq 8\eta + 2\sum_{i=1}^{K} \frac{\eta w_t^2(i)}{w_t(i) + \gamma\eta}\left(z_t^2(i)\widehat{\ell}_t^2(i) + \eta^2 z_t^2(i)\widehat{\ell}_t^4(i)\right)$$

$$\leq 8\eta + 2\eta\sum_i w_t(i)z_t(i)\widehat{\ell}_t^2(i) + \frac{2\eta^2}{\gamma}\sum_i w_t^2(i)z_t^2(i)\widehat{\ell}_t^4(i).$$

Now we use the fact $\mathbb{E}_t\left[\widehat{\ell}_t^2(i)\right] \leq \frac{\ell_t^2(i)}{\tilde{p}_t(i)} \leq \frac{2\ell_t^2(i)}{p_t(i)}$ and $\mathbb{E}_t\left[\widehat{\ell}_t^4(i)\right] \leq \frac{\ell_t^2(i)}{\tilde{p}_t^3(i)} \leq \frac{4K\ell_t^2(i)}{\eta p_t^2(i)}$ to continue with

$$\mathbb{E}_t\left[\|c_t\|^2_{\nabla^{-2}\psi(w_t)}\right] \leq 8\eta + 4\eta\sum_i \ell_t^2(i) + \frac{8\eta K}{\gamma}\sum_i \ell_t^2(i) = \mathcal{O}(\rho\eta).$$

This finishes the proof. $\qquad\qquad\qquad\qquad\qquad\qquad\qquad\qquad\qquad\qquad\qquad\square$

*Proof of Lemma 14.* By the definition of $z_{t+1}(i)$ and first order optimality, one has

$$(u_t - z_{t+1}(i))(-r_t(i) + \phi'(z_{t+1}(i)) - \phi'(z_t(i))) \geq 0,$$

which after rearranging gives

$$-(z_{t+1}(i) - u_t)r_t(i) \leq (u_t - z_{t+1}(i))(\phi'(z_{t+1}(i)) - \phi'(z_t(i)))$$
$$= D_\phi(u_t, z_t(i)) - D_\phi(u_t, z_{t+1}(i)) - D_\phi(z_{t+1}(i), z_t(i))$$
$$\leqslant D_\phi(u_t, z_t(i)) - D_\phi(u_t, z_{t+1}(i)).$$

Summing over $t$, telescoping, and realizing $D_\phi(u_t, z_t(i)) = \frac{1}{\eta}\left(\frac{u_t}{z_t(i)} + \ln\frac{z_t(i)}{u_t} - 1\right) \leq \frac{2}{\eta\delta}$ since $u_t$ and $z_t(i)$ are in $[\delta, 1]$, we arrive at

$$-\sum_{t=1}^{T} z_{t+1}(i)r_t(i) + \sum_{t=1}^{T} u_t r_t(i) \leq \frac{2S_i}{\eta\delta}.$$

It remains to prove $(z_{t+1}(i) - z_t(i))r_{t,i} \leq \eta\sum_t z_t(i)r_t^2(i)$. For notational convenience, given any $L, \xi \in \mathbb{R}$, let $z_1 = \operatorname{argmin}_{z\in[\delta,1]} Lz + \phi(z)$ and $z_2 = \operatorname{argmin}_{z\in[\delta,1]}(L+\xi)z + \phi(z)$. If we can prove $\xi(z_1 - z_2) \leq \eta z_1\xi^2$, then we finish the proof by setting $L = -\phi'(z_t(i))$ and $\xi = -r_t(i)$ (which gives $z_1 = z_t(i)$ and $z_2 = z_{t+1}(i)$).

To show $\xi(z_1 - z_2) \leq \eta z_1\xi^2$. Realize that the optimizations are one dimensional and admit the following solutions with explicit forms

$$z_1 = \begin{cases} 1 & \text{if } L \leq \frac{1}{\eta} \\ \frac{1}{\eta L} & \text{if } \frac{1}{\eta} < L < \frac{1}{\eta\delta} \\ \delta & \text{if } L \geq \frac{1}{\eta\delta} \end{cases}, \quad z_2 = \begin{cases} 1 & \text{if } L+\xi \leq \frac{1}{\eta} \\ \frac{1}{\eta(L+\xi)} & \text{if } \frac{1}{\eta} < L+\xi < \frac{1}{\eta\delta} \\ \delta & \text{if } L+\xi \geq \frac{1}{\eta\delta} \end{cases}$$

The rest of the proof is simply to show $\xi(z_1 - z_2) \leq \eta z_1\xi^2$ holds in all of the nine possible cases.

    A. If $z_1 = z_2 = 1$, then $\xi(z_1 - z_2) = 0 \leq \eta z_1\xi^2$ holds trivially.

B. If $z_1 = 1$ and $z_2 = \frac{1}{\eta(L+\xi)}$, then $L - \frac{1}{\eta} \leq 0$ and $\eta(L + \eta) \geq 1$ and thus

$$\xi(z_1 - z_2) = \eta\xi\frac{L + \xi - 1/\eta}{\eta(L + \xi)} \leq \eta\xi^2 = \eta z_1\xi^2.$$

C. If $z_1 = 1$ and $z_2 = \delta$, then $\xi \geq 0$, $L \leq \frac{1}{\eta}$, and $\frac{1}{\eta\delta} - L \leq \xi$, and thus

$$\xi(z_1 - z_2) = \xi(1 - \delta) \leq \xi\frac{1 - \delta}{\delta} = \eta\xi\left(\frac{1}{\eta\delta} - \frac{1}{\eta}\right)$$

$$\leq \eta\xi\left(\frac{1}{\eta\delta} - L\right) \leq \eta\xi^2 = \eta z_1\xi^2.$$

D. If $z_1 = \frac{1}{\eta L}$ and $z_2 = 1$, then $\xi \leq 0$ and $\eta L - 1 \leq -\eta\xi$, and thus

$$\xi(z_1 - z_2) = z_1|\xi|(\eta L - 1) \leq \eta z_1\xi^2.$$

E. If $z_1 = \frac{1}{\eta L}$ and $z_2 = \frac{1}{\eta(L+\xi)}$, then $\frac{1}{L+\xi} \leq \eta$, and thus

$$\xi(z_1 - z_2) = \frac{z_1\xi^2}{L + \xi} \leq \eta z_1\xi^2.$$

F. If $z_1 = \frac{1}{\eta L}$ and $z_2 = \delta$, then $\xi \geq 0$ and $\frac{1}{\eta\delta} - L \leq \xi$, and thus

$$\xi(z_1 - z_2) = \eta z_1\xi\delta\left(\frac{1}{\eta\delta} - L\right) \leq \eta z_1\xi^2.$$

G. If $z_1 = \delta$ and $z_2 = 1$, then $\xi \leq 0$, $\frac{1}{\eta\delta} \leq L$, and $L - \frac{1}{\eta} \leq \xi$, and thus

$$\xi(z_1 - z_2) = \eta z_1|\xi|\left(\frac{1}{\eta\delta} - \frac{1}{\eta}\right) \leq \eta z_1|\xi|\left(L - \frac{1}{\eta}\right) \leq \eta z_1\xi^2.$$

H. If $z_1 = \delta$ and $z_2 = \frac{1}{\eta(L+\xi)}$, then $\xi \leq 0$, $\frac{1}{\eta L} \leq \delta$, $\frac{1}{\eta(L+\xi)} \leq 1$, and thus

$$\xi(z_1 - z_2) \leq |\xi|\left(z_2 - \frac{1}{\eta L}\right) = \frac{\xi^2}{\eta L(L + \xi)} \leq \frac{\xi^2}{L} \leq \eta\delta\xi^2 = \eta z_1\xi^2.$$

I. If $z_1 = z_2 = \delta$, then $\xi(z_1 - z_2) = 0 \leq \eta z_1\xi^2$ holds trivially.

This finishes the proof. □