[Reviews · NeurIPS 2019]

Reviewer 1



Summary: This paper proposes black-box reduction methods from online learning with experts to that of tracking best shifting experts. In particular, the paper considers two settings, full information, and adversarial bandit setting, where the latter gives only feedbacks of chosen experts. The proposed algorithms achieve better bounds than previous work, as well as adaptive algorithms. For the bandit tracking problems, the paper also shows a lower bound of the regret. Comments: The proposed black-box reduction from tracking experts to competing with a static optimal expert is new (as far as I know) and a breakthrough in the theory. The algorithms are new but simple. The results looks feasible while I have not checked the details. The technical contribution is substantial and I recommend this paper for acceptance.

Reviewer 2



The paper is nice to read, the proofs are sound and the literature seem to be well cited. On the negative side, the different regret bounds proved in this paper improve existing regret bounds in very specific setting (depending on n, S, and the total number of experts) and some of them may not be optimal (such as the one on bandits with sparsity). This might thus only interest a small niche of experts only. Other comments: - Some discussion should be made somewhere about the optimality of the bounds. What is known to be optimal and what is not? - About the bandit feedback with sparsity: it would be nice to add figures that compare the rates of the existing bound and the one of Thm. 8 to better understand when this one is better and when it is not. In particular, it is stated that this is the case when S and K are very large. But only S needs to be large, since the assumption $T>S>max{T/K^3,K^5/T}$ is enough which can be satisfied for small K. - About the existing results for the stochastic setting (see line 47-52): when reading this paragraph, I am not sure if the existing results refer to best-of-both-worlds results only or to any results for switching regret in stochastic environment. Does there exists any results for tracking a small set of experts for stochastic losses? - I am wondering if an algorithm such as Coral (see Agarwal, Luo et al. 2016) could be use in the sparse bandit setting to improve the rate to sqrt{T} instead of T^{2/3}.

Reviewer 3



The main novelty in the full-information part of the results is the black-box reduction to the confidence-based framework. The main intuitive idea is that confidences are generated for experts based on how good they have looked in the past/how bad regret has been with respect to them so far (Step 7 of Algorithm 1), and the confidence is multiplied by the probabilities generated by the actual expert algorithm *with a static regret guarantee*. The algorithm and analysis provides a conceptual look into how switching regret minimization can be achieved and is interesting. Proof (of Theorem 3) appears essentially correct. These ideas also yield new results for the sparse bandit version of the problem, which has seen recent progress. I did not check the proofs in detail, but it seems that the building blocks seem a bit easier to achieve than the original expert-based approach (here, they are achieving worst-case static regret, and worst-case switching regret over 2 actions, as opposed to directly achieving worst-case switching regret over K actions). It still requires sophisticated regularization with OMD "one-sided log barrier", which I have not seen in prior work and appears to be a technical contribution. Proofs appear correct (although I only checked in detail for full-information) and formally written. The submission is also nicely contextualized in related work. While the results are nice, I am not sure about the broad appeal of this problem to people in the NeurIPS community outside of the online learning community; so it would be nice to hear from the authors on whether they think the regret guarantee and this problem more generally has scope in any practical machine learning application.

[Author Response · NeurIPS 2019]

We thank all reviewers for their valuable comments. Before we answer specific questions raised by the reviewers, we would like to first address the common concern on the broad appeal of our results outside the online learning community and its practical value. While our results are stated for two particular online learning problems (expert problem and multi-armed bandits), it is worth pointing out that 1) these are two fundamental learning problems and have numerous applications in both theory and practice; 2) our black-box approach provides a much more intuitive understanding of the problem and also gives an easy way to design algorithms with long-term memory, which we believe could be applied to other problems; 3) algorithms with long-term memory have been applied to practical applications such as TCP round-trip time estimation [4], intrusion detection system [3], and multi-agent systems [5], and we believe that our algorithms (especially the adaptive one) could potentially lead to better practical performance.

**Reviewer 2:**

— "the different regret bounds proved in this paper improve existing regret bounds in very specific setting":
This is admittedly true from a theoretical viewpoint. However, it is worth noting that algorithms with long-term memory indeed often exhibit superior empirical performance than those without, as shown in previous works (such as [1, 2]). Therefore, we believe that the significance of our results goes beyond the theoretical improvement of regret bounds.

— "Some discussion should be made somewhere about the optimality of the bounds."
We will add more discussion on this in the next version of our paper, as suggested by the reviewer. For the full information setting, as far as we know there is no existing lower bound. Note that, however, our upper bound (and that of [1]) essentially matches the bound of the computationally inefficient approach of running Hedge over all sequences with $S$ switches among $n$ experts, an approach that usually leads to the information-theoretically optimal regret bound. For the bandit setting, again there is no known lower bound. We do not believe that our bound is optimal and characterizing the optimal regret in this case is left as a future direction.

— "it would be nice to add figures that compare the rates of the existing bound and the one of Thm. 8"
We thank the reviewer for this suggestion. We will add this to the next version of our paper.

—"About the existing results for the stochastic setting (see line 47-52): ...":
The existing results refer to any results for switching regret in the stochastic environment. We are not aware of any existing results for tracking a small set of experts with stochastic losses. The problem is explicitly stated as an open problem in [6].

— Whether Corral algorithm can be used to improve the results for the sparse bandit setting:
We indeed have thought about this carefully, but in short we could not make it work. Note that there are two important differences here compared to the Corral setup: 1) we need to avoid polynomial dependence on $K$ (except for lower-order terms) and 2) we need to have switching regret bound (instead of static regret, as in Corral) for the sub-routines.

**Reviewer 3:**

— "It is worth noting that none of the full-information results are *new* by themselves. ... AdaNormalHedge.TV gets similar guarantees in the stochastic setting although suboptimal in log factors"
We respectfully disagree with this comment. Our best-of-both-worlds result (or even just the result for the stochastic case) is new and resolves the open problem of Koolen and Warmuth [6]. What AdaNormalHedge.TV achieves is the typical switching regret bound, involving a term $S \ln T + S \ln K$ (for either adversarial or stochastic setting), while our results improve this term to $S \ln T + n \ln K$ (not just log factors), which is the typical and desirable improvement for this problem and is meaningful for large $K$ (for example, the first paper on this topic by Bousquet and Warmuth [1] obtains the exact same improvement, but only in the adversarial case).

# References

[1] O. Bousquet and M. K. Warmuth. Tracking a small set of experts by mixing past posteriors. *Journal of Machine Learning Research*, 3(Nov):363–396, 2002.

[2] R. B. Gramacy, M. K. Warmuth, S. A. Brandt, and I. Ari. Adaptive caching by refetching. In *Advances in Neural Information Processing Systems*, pages 1489–1496, 2003.

[3] H. T. Nguyen and K. Franke. Adaptive intrusion detection system via online machine learning. In *2012 12th International Conference on Hybrid Intelligent Systems (HIS)*, pages 271–277. IEEE, 2012.

[4] B. A. A. Nunes, K. Veenstra, W. Ballenthin, S. Lukin, and K. Obraczka. A machine learning framework for tcp round-trip time estimation. *EURASIP Journal on Wireless Communications and Networking*, 2014(1):47, 2014.

[5] T. Santarra. *Communicating Plans in Ad Hoc Multiagent Teams*. PhD thesis, UC Santa Cruz, 2019.

[6] M. K. Warmuth and W. M. Koolen. Open problem: Shifting experts on easy data. In *Conference on Learning Theory*, pages 1295–1298, 2014.


[Meta-Review · NeurIPS 2019]

This paper makes revisits the long-term memory problem (switching within a small set of experts) in the full info and bandit cases, with specific focus on obtaining improved results in the stochastic case. The reviewers are all appreciative of the new black-box reduction proposed in this paper, and the way in which it is applied to the sparse bandit problem. Of course it would be nice to have lower bounds and experiments... Yet the consensus is that the paper makes a sizable contribution as-is. During the review the question was raised whether this arguably niche problem would find sufficient interest at NeurIPS. Past instances of the conference did host papers discussing aspects of this probem, see e.g. references [2], [13] and [34]. It appears that worry about this is unfounded.